

# Carbon export and burial pathways driven by a low-latitude arc-continent collision

Amy I. Hsieh[1], Thierry Adatte[1], Shraddha Band[2], Li Lo[3], Romain Vaucher[4], Brahimsamba Bomou[1], Laszlo Kocsis[5], Pei-Ling Wang[6], Samuel Jaccard[1]

[1]Institute of Earth Sciences, University of Lausanne, Lausanne, CH-1015, Switzerland
[2]Biodiversity Research Center, Academia Sinica, Taipei, 115, Taiwan
[3]Department of Geosciences, National Taiwan University, Taipei, 106, Taiwan
[4]College of Science and Engineering, James Cook University, Townsville, 4814, Australia
[5]Institute of Earth Surface Dynamics, University of Lausanne, Lausanne, CH-1015, Switzerland
[6]Institute of Oceanography, National Taiwan University, Taipei, 106, Taiwan

*Correspondence to*: Amy I. Hsieh (hsiehiamy@gmail.com)

**Abstract.** Chemical weathering of silicate rocks of low-latitude arc–continent collisions has been hypothesized as a driver of global cooling since the Neogene. In low-latitude regions, monsoon and tropical cyclone precipitation also drive intense physical erosion that contribute to terrestrial carbon export and nutrient-stimulated marine productivity. Despite this, the role of physical weathering on carbon sequestration has largely been overlooked. To address this gap, we analyse late Miocene–early Pleistocene sedimentary and geochemical records from the Taiwan Western Foreland Basin and time-equivalent records from the northern South China Sea.

Along the continental slope, organic carbon is largely marine in origin, and its accumulation controlled by long-term sea-level fall and glaciation. In contrast, on the continental rise, organic carbon burial is controlled by high sedimentation rates related to Taiwan's uplift and erosion (since ~5.4 Ma). Despite increased terrestrial erosion of Taiwan, the organic material remains mainly marine in origin, suggesting that primary production was enhanced by nutrient exported from Taiwan. Marine organic matter along Taiwan's shore was subsequently remobilized by turbidity currents through submarine canyon systems and accumulating on the continental rise of Eurasia. The onset of Northern Hemisphere Glaciation (~3 Ma) and subsequent intensification of the East Asian Summer Monsoon and persistent tropical cyclone activity all further amplified nutrient export across the basin, further stimulating marine primary production.

Our findings demonstrate that arc–continent collision influences carbon sequestration through two pathways: (1) direct burial of terrestrial organic matter and (2) nutrient-fuelled marine productivity and burial. This work establishes a direct link between the erosion of an arc-continent collision and long-term carbon burial in adjacent ocean basins.

## 1 Introduction

Global cooling since the late Eocene has traditionally been attributed to tectonic forcing and enhanced chemical weathering of silicate rock from the Himalayan and Tibetan Plateau (Raymo and Ruddiman, 1992), which results in the removal of atmospheric $CO_2$ (Walker et al., 1981). However, weathering fluxes have decreased in both regions during the Neogene (Clift and Jonell, 2021), and global silicate fluxes appear to have remained near steady-state through the Cenozoic (Caves et al., 2016) even as global cooling continued. To reconcile stable or declining chemical



weathering rates with decreasing atmospheric $CO_2$, an alternative hypothesis emphasized chemical erosion of arc-
continent collisional orogens in low-latitude, tropical regions (Bayon et al., 2023; Clift et al., 2024; Jagoutz et al.,
2016; Macdonald et al., 2019). In such environments, warm and humid conditions amplify chemical weathering,
enhancing carbon removal and sequestration. While existing studies support a correlation between the growth and
weathering of low-latitude orogens and long-term atmospheric $CO_2$ concentration and global temperature records,
they have yet to fully account for the roles of physical erosion, terrestrial organic carbon burial, and changes in marine
productivity.
In low-latitude regions, tropical cyclones and monsoons are the primary drivers of erosion and sediment dispersal,
delivering elevated sediment loads to adjacent seas via intense precipitation and high river discharge from steep
mountainous catchments (Chen et al., 2018; Milliman and Kao, 2005). Warm sea-surface temperatures and reduced
polar ice volumes under past greenhouse climates likely amplified monsoon variability and produced tropical cyclones
that were considerably more intense and frequent than at present (Fedorov et al., 2013). These conditions of elevated
humidity and precipitation would have promoted not only enhanced chemical weathering of silicate rocks, but also
greater terrestrial biomass production.
Land-to-sea export of terrestrial organic material from vegetation, soil, and rock is enhanced under high precipitation
regimes, with steep mountain rivers efficiently transporting this material for burial in adjacent ocean basins (Hilton et
al., 2011; Milliman et al., 2017). The global terrestrial carbon pool accounts for ~7.5% of the Earth's total carbon
stock, excluding lithospheric carbon, and is more than five times larger than the atmospheric carbon pool (Canadell et
al., 2021). As a result, even modest changes in the terrestrial carbon storage can significantly alter atmospheric $CO_2$
concentrations (Houghton, 2003). In particular, physical erosion by water is widely recognized as a dominant control
of land–atmosphere carbon exchange (Hilton and West, 2020; Van Oost et al., 2012). Elevated sediment discharge to
the oceans would facilitate the export and burial of terrestrial organic carbon (Aumont et al., 2001; Dagg et al., 2004;
Galy et al., 2007; Hilton et al., 2011; Jin et al., 2023; Liu et al., 2013), and also deliver bioessential nutrients that
stimulate marine productivity (Beusen et al., 2016; Dürr et al., 2011; Hoshiba and Yamanaka, 2013; Krumins et al.,
2013). However, the role of fluvial nutrient export in fueling marine primary productivity is generally thought to be
limited to coastal regions (Dagg et al., 2004; Froelich, 1988; Stepanauskas et al., 2002). This oversimplification in
ocean biogeochemical models leads to a poorly constrained link between terrestrial nutrient supply, open-ocean
productivity, and deep-sea carbon burial.
This research aims to address these knowledge gaps by disentangling the different mechanisms through which carbon
is sequestered as a result of low-latitude arc-continent collisions (Fig. 1). A clearer understanding of these processes
will provide stronger constraints on both reconstructed and predictive carbon budget models. The study area focuses
on the northern South China Sea (SCS) region, specifically late Miocene to early Pleistocene (~6.3–2 Ma) strata of
the Taiwan Western Foreland Basin (TWFB, i.e., paleo-Taiwan Strait; Fig. 2) and time-equivalent sediment core
records obtained from the Ocean Drilling Program (ODP Sites 1146 and 1148; Fig. 2). Since its emergence in the
early Pliocene, Taiwan has been characterized by exceptionally high denudation rates and rapidly became the
dominant sediment source to the adjacent TWFB, overwhelming contributions from southeast Eurasia (Hsieh et al.,
2023b). Hyperpycnal flows triggered by intense precipitation transported Taiwan-derived sediments over 1000 km



into the SCS, leaving a distinct signature in deep-sea deposits (Hsieh et al., 2024; Liu et al., 2012). Strata of the TWFB
capture the evolution of the Taiwan Orogen (Lin and Watts, 2002), and thus provide insight into how changes in
weathering and erosion processes modulated carbon burial in the SCS sediments across successive orogenesis stages.

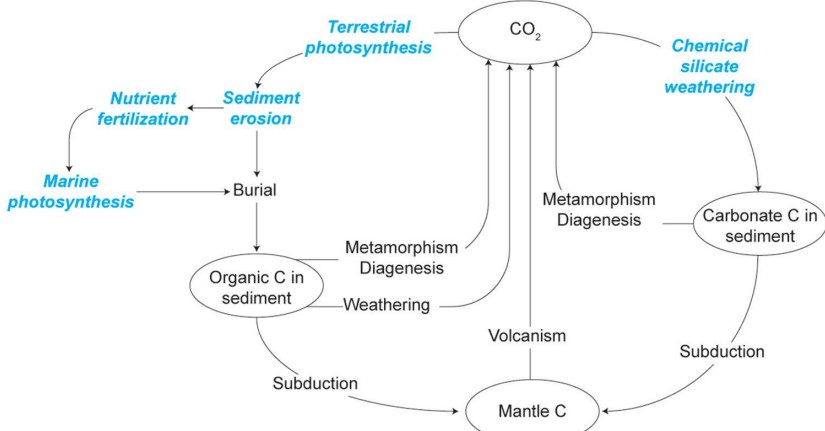


**Figure 1: Conceptual model of geologic carbon (C) sources and sinks, modified from (Berner, 2003). This research focuses**
**on two main pathways of carbon sequestration often associated with arc-continent collisions, highlighted in blue: (1) direct**
**burial of terrestrial organic matter, and (2) nutrient-fueled marine productivity followed by the burial of marine organic**
**matter. These processes play a crucial role in the long-term carbon cycle and the regulation of atmospheric CO₂.**

## 2 Study area

The base of the TWFB stratigraphic fill is composed of the Kueichulin Formation (Fm; late Miocene–early Pliocene),
a sandstone-dominated unit deposited in shallow-marine and deltaic environments under the influence of wave and
tidal processes, and composed of three members (from base to top): the Kuantaoshan Sandstone, Shihliufen Shale,
and Yutengping Sandstone (Fig. 2; Castelltort et al., 2011; Hsieh et al., 2025; Nagel et al., 2013). Overlying the
Kueichulin Fm is the Chinshui Shale (late Pliocene), a mudstone-rich succession with uncommon wavy-laminated
sandstone interbeds that accumulated in an offshore setting during a phase of maximum flooding and enhanced
subsidence in the TWFB (Castelltort et al., 2011; Nagel et al., 2013; Pan et al., 2015). The Chinshui Shale is overlain
by the Cholan Fm (early Pleistocene), which consists of heterolithic sediments deposited in shallow-marine
environments influenced by waves, rivers, and tides (Covey, 1986; Nagel et al., 2013; Pan et al., 2015; Vaucher et al.,
2023a).
The targeted time interval (~6.27–1.95 Ma) spans the initiation of Eurasian-Philippine plate collision through the
mergence and uplift of Taiwan. It includes the Pliocene (5.33–2.58 Ma), which may be the most recent time in Earth's
history when atmospheric $CO_2$ last reached or exceeded present-day concentrations (>400 ppm; Tierney et al., 2019),
and the subsequent transition toward Pleistocene icehouse conditions. Since tectonic configurations, insolation, and
major floral and faunal assemblages have remained broadly unchanged since the mid-Pliocene (Dowsett, 2007;
Robinson et al., 2008), this period provides a critical Earth system analog for evaluating future climate hazards (e.g.,
Burke et al., 2018), including sea-level rise and extreme weather events.

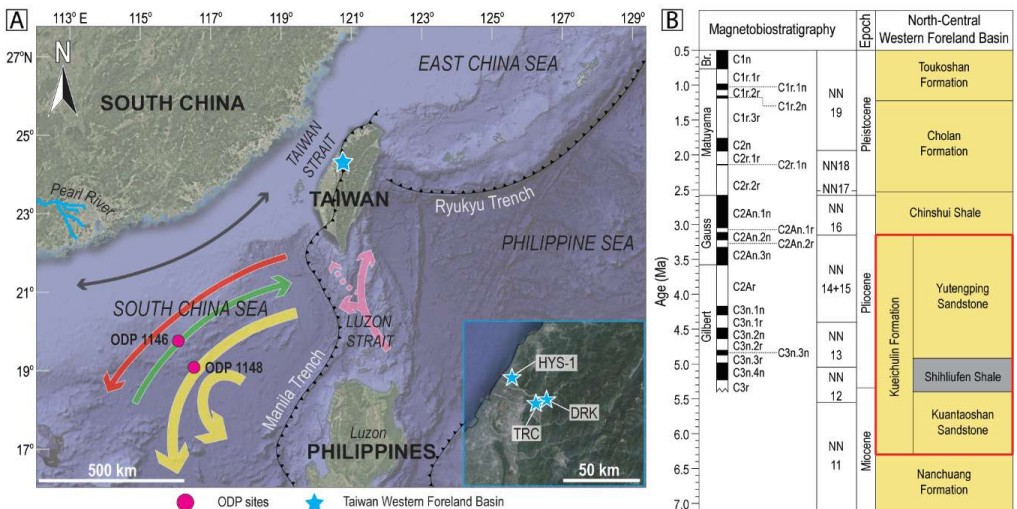


**Figure 2: A) Map of the study area showing the locations of the Late Miocene–Early Pleistocene records from Ocean Drilling Program (ODP) sediment cores in the South China Sea, and the outcrop of the Kueichulin Fm from the Taiwan Western Foreland Basin (TWFB). The inset map outlined in blue show the locations of the borehole (HYS-1) and outcrop locations (DRK = Da'an River, Kueichulin Fm; TRC = Tachia River, Chinshui Shale) of the TWFB strata used in this study. Modern-day circulation in the SCS is shown in arrows: black = alongshore surface current, red = surface water current, green = intermediate water current, yellow = deep- and bottom-water current, pink = Kuroshio current, pink (dashed) = Taiwan warm current (modified from Hu et al. (2010); Liu et al. (2010a); Liu et al. (2016); Yin et al. (2023)). B) Chronostratigraphy of the TWFB is modified after Chen (2016), Hsieh et al. (2023a), and Teng et al. (1991). The red box highlights the targeted study section. Yellow denotes sandstone, and grey indicates mudstone.**

**3 Methodology**
**3.1 Data acquisition and analysis**
A total of 553 samples were collected from outcrops of the TWFB exposed along rivers in southwestern Taiwan,
including 272 collected from the Kueichulin Fm by Dashtgard et al. (2021) and Hsieh et al. (2023b) along the Da'an
River. This was combined with new data from the Chinshui Shale (n=90; Tachia River) and the Cholan Fm (n=191;
Houlong River). Data between 4.13–3.15 Ma are not available as no outcrop sections were accessible. Gamma-ray
data were obtained from the HYS-1 borehole drilled through the TWFB. Age-equivalent material was also obtained
from deep-sea sediment cores ODP Site 1146 (19°27.40'N, 116°16.37'E, 2092 m water depth, 179.8–343.1 m core
depth; Holbourn et al., 2005; Holbourn et al., 2007) and Site 1148 (18°50.169'N, 116°33.939'E, 3294 m water depth,
118.9–206 m core depth; Cheng et al., 2004; Tian et al., 2008), archived in international core repositories. Sampling
resolution averaged ~1.4 m vertically through the TWFB stratigraphic sections, and ~0.65 m and ~0.35 m through the
ODP Sites 1146 and 1148 cores, respectively.
Samples from the Chinshui Shale and ODP sites were analysed for organic geochemistry and paleomagnetism. For
the Chinshui Shale, total organic carbon (TOC) and total nitrogen (TN) concentrations were determined from
pulverized rock samples in the Department of Geosciences at National Taiwan University (NTU) using an elemental
analyser (Elementar TOC analyser soli TOC® cube; Lin et al., 2025). Total carbon (TC) and TN abundances for ODP
samples were determined with a CHNS Elemental Analyser (Thermo Finnigan Flash EA 1112) at the Institute of Earth



Sciences (ISTE) at the University of Lausanne in Switzerland on oven-dried sieved and crushed sediment samples.
The samples were heated to 900°C, after which the combustion products were extracted into a chromatographic
column where they were converted into simpler components: $CO_2$ and $N_2$. These components were then measured by
a thermal conductivity detector, and the results were expressed as a weight percentage. Analytical precision and
accuracy were determined by replicate analyses and by comparison with an organic analytical standard composed of
purified L-cysteine, achieving a precision of better than 0.3% (REFS). Organic matter (OM) analyses of ODP core
samples were performed on whole-rock powdered samples using a Rock-Eval 6 at the ISTE following the method
described by Espitalie et al. (1985) and Behar et al. (2001). Measurements were calibrated using the IFP 160000
standard. Rock-Eval pyrolysis provides parameters such as hydrogen index (HI, mg HC $g^{-1}$ TOC, HC = hydrocarbons),
oxygen index (OI, mg $CO_2$ $g^{-1}$ TOC), $T_{max}$ (°C), and the TOC (wt.%). HI, OI and $T_{max}$ values, which give an overall
measure of the type and maturation of the organic matter (e.g., Espitalie et al., 1985), can't be interpreted for TOC <
0.2 wt.% and $S_2$ values $\geq$ 0.2 mg HC $g^{-1}$. Total organic carbon accumulation rates (mg $cm^{-2}$ $kyr^{-1}$) for the ODP sites
were calculated by multiplying mass-accumulation rates (MAR) derived from literature and TOC.
Organic carbon isotopic compositions ($\delta^{13}C_{org}$, ‰ relative to Vienna Pee Dee Belemnite) were measured by flash
combustion on an elemental analyser (EA) coupled to an isotope-ratio mass spectrometer (IRMS) from pulverized,
decarbonated (10% HCl treatment) whole-rock samples. Samples from ODP sites were analysed at the Institute of
Earth Surface Dynamics, University of Lausanne, using a Thermo EA IsoLink CN connected to a Delta V Plus isotope
ratio mass spectrometer (Thermo Fisher Scientific, Bremen), both operated under continuous helium flow. The
samples and standards are weighed in tin capsules and combusted at 1020°C with oxygen pulse in a quartz reactor
filled with chromium oxide ($Cr_2O_3$) and below with silvered cobaltous-cobaltic oxide. The combustion produced gases
($CO_2$, $N_2$, $NO_x$ and $H_2O$) are carried by the He-flow to a second reactor filled with elemental copper and copper oxide
wires kept at 640°C to remove excess oxygen and reduce non-stoichiometric nitrous products to $N_2$. The gases are
then carried through a water trap filled with magnesium perchlorate ($Mg(ClO_4)$). The dried $N_2$ and $CO_2$ gases are
separated with a gas chromatograph column at 70 °C and then carried to the mass spectrometer. The measured $\delta^{13}C$
values are calibrated and normalized using international reference materials and in-house standards Spangenberg,
2016. Samples from the Chinshui Shale were analysed at the Stable Isotope Laboratory at National Taiwan University
using a Flash EA (Thermo Fisher Scientific) coupled to a Delta V Advantage (Thermo Fisher Scientific). The $\delta^{13}C$
values are calibrated using an international reference material, IAEA-CH-3. The reproducibility and accuracy are
better than ±0.1‰.
Thirty-three oriented palaeomagnetic core specimens (25-mm diameter) were collected at ~3.5 m intervals from
unweathered, mud-rich beds, then prepared and analysed at Academia Sinica in Taiwan following the methodology
described in Horng (2014). Cores were cut into 2-cm samples, and bulk magnetic susceptibility measured using a
Bartington Instruments MS2B magnetic susceptibility meter. Mass-specific magnetic susceptibility ($\chi$) was then
derived by normalising bulk magnetic susceptibility to sample mass.
Existing data for the ODP sites 1146 and 1148 were also compiled from literature, including clastic MAR (Site 1146
from Wan et al., 2010a, Site 1148 from Wang et al., 2000a), magnetic susceptibility (1146 from Wang et al., 2005a,
1148 from Wang et al., 2000a), hematite/goethite ratios (Hm/Gt) derived from spectral reflectance band ratios at



565/435 nm (1146 from Wang et al., 2000b, 1148 from Clift, 2006), continuous gamma-ray logs (1146 from Wang et
al., 2000b, 1148 from Wang et al., 2000a), and titanium/calcium ratios (Ti/Ca; 1146 from Wan et al., 2010a, 1148
from Hoang et al., 2010). MAR, magnetic susceptibility, and Ti/Ca serve as proxies for physical erosion, recording
variations in terrigenous sediment flux linked to summer monsoon precipitation. Intensified precipitation enhances
basin sediment accumulation rates (Clift et al., 2014), and typically increases the magnetic susceptibility of marine
sediment via enhanced runoff and terrestrial input (Clift et al., 2002; Kissel et al., 2017; Tian et al., 2005). In the SCS,
magnetic susceptibility also serves as a sediment provenance indicator. Sediment sourced from western Taiwan yields
$\chi$ values that range from $0.9 \pm 0.3$ to $1.8 \pm 0.5 \times 10^{-7}$ m$^3$ kg$^{-1}$, much lower than those sourced from the South China
Block ($4.0 \pm 1.3 \times 10^{-7}$ m$^3$ kg$^{-1}$), indicating a relative depletion of magnetic minerals in Taiwan-sourced material (Horng
and Huh, 2011). Titanium, associated with heavy mineral deposition, and calcium, linked to pelagic biogenic
carbonate accumulation, yield Ti/Ca values that increase with enhanced monsoon-driven sediment export (Clift et al.,
2014). Gamma-ray intensities broadly track changes in lithology (Green and Fearon, 1940; Schlumberger, 1989),
where values < 75 American Petroleum Institute (API) typically mark sandstone-rich intervals, > 105 API mudstone-
rich intervals, and intermediate values reflect mixed lithologies. Increased sediment export, particularly of coarser
grains, may be expressed as lower API values.
Sedimentary TOC content provides a measure of organic carbon accumulation through time. Terrestrial and marine
sources can also be differentiated by their $\delta^{13}C_{org}$ values (Chmura and Aharon, 1995; Dashtgard et al., 2021; Hilton et
al., 2010; Martiny et al., 2013; Peterson and Fry, 1987). Marine organic matter (e.g., plankton, particulate and
dissolved organic matter) typically have more enriched values than terrestrial inputs (e.g., C3 and C4 plants, and soil
and lithogenic organic carbon) (Table 1). Marine-derived organic matter mainly accumulates on the seafloor under
fair-weather conditions, while terrestrial input increases under intervals of increased precipitation and erosion
(Dashtgard et al., 2021; Hsieh et al., 2023b).
**Table 1: Typical values for marine- and terrestrially sourced $\delta^{13}C_{org}$ and C/N (compiled by Dashtgard et al., 2021). Numbers**
**in brackets represent sample count. OM = organic material.**

| | Organic Material | $\delta^{13}C_{org}$ (‰) | C/N |
|---|---|---|---|
| Marine | Particulate OM | -22.5 ± 1.7 (53) | 6.2 ± 1.0 |
| | Plankton | -20.4 ± 1.4 (184) | - |
| | Dissolved OM | -22.5 ± 0.8 (23) | - |
| | All pelagic marine organic matter - equally weighted | -21.8 ± 1.7 | 6.2 ± 1.0 |
| Terrestrial | High-$^{13}$C plants (C4) | -13.2 ± 1.9 (89) | 83.3 ± 54 (6) |
| | Low-$^{13}$C plants (C3) | -27.4 ± 1.9 (161) | 52 ± 14.8 (55) |
| | Soil | -25.9 ± 1.2 (11) | 17.1 ± 7.3 (22) |






Hematite-to-goethite (Hm/Gt) ratios are widely applied as indicator of monsoon precipitation (Clift, 2006; Liu et al.,
2007; Zhang et al., 2009). Hematite typically forms through iron oxidation under arid climates, whereas goethite
preferentially develops under humid climates (e.g., Kämpf and Schwertmann, 1983; Maher, 1986). In the northern
SCS, however, Clift et al. (2014) documented a positive relationship between elevated Hm/Gt values and intensified
East Asian Summer Monsoon (EASM) rainfall and seasonality. Beyond climate, hematite also reflects sediment
provenance: sediment derived from Taiwan is notably depleted in hematite and enriched in pyrrhotite (Horng and
Huh, 2011). Locally estimated scatterplot smoothing (LOESS) is applied to all data to reveal trends through the studied
time interval (Cleveland et al., 1992).

**3.2 Age models**

The chronostratigraphic framework for the Kueichulin Fm, Chinshui Shale, and and Cholan Fm of the TWFB was
established by astronomically tuning the gamma-ray records to the $\delta^{18}O$ record of Wilkens et al. (2017) (Hsieh et al.,
2023a; Vaucher et al., 2023b). However, the boundary between the top of the Kueichulin Fm and the base of the
Chinshui Shale is not well-established. Therefore, a magnetobiostratigraphic age model was developed from
nannofossil zones and magnetic reversals identified in oriented outcrop core samples from the Chinshui Shale outcrop
using the methodology described in Horng (2014) to ground-truth the existing framework. The remanent magnetic
intensity, and declination and inclination of oriented core samples were measured using a JR-6A spinner
magnetometer (AGICO). To determine the stable remanent magnetization and polarity (i.e., normal or reversed) of
each sample, unstable secondary magnetization was removed by thermally demagnetizing the samples stepwise from
25 to 600°C. The characteristic remanent magnetization (ChRM) declination and inclination of thermally
demagnetized samples were calculated using principal component analysis with a minimum of three demagnetization
steps in the PuffinPlot software (Lurcock and Wilson, 2012) to determine the polarity of each sample. Thermal
demagnetization diagrams for the Chinshui Shale samples showing the stable remanent magnetic declinations and
inclinations after principal component analysis are presented in Fig. S1 in Supporting Information.
Index nannofossils and corresponding biozonations identified by Shea and Huang (2003) for the Chinshui Shale were
used to constrain paleomagnetic polarities. The resulting age model was then correlated to an orbitally tuned, benthic
foraminiferal, stable oxygen isotope ($\delta^{18}O$) record from the equatorial Atlantic Ocean (Wilkens et al., 2017), which is
tied to physical sedimentary properties independent of ice volume, and has a robust timescale. Variations in both
parameters are assumed to be causally linked and temporally in phase.
The age model for ODP Site 1146 (Wan et al., 2010a) was constructed by linear interpolation between
magnetobiostratigraphic age control points established by Wang et al. (2000b). Stratal ages from ODP Site 1148 (Clift,
2006) are constrained using biostratigraphic ages of benthic foraminifera (Wang et al., 2000a).

**4 Results**

Data collected from the Chinshui Shale (n = 90) for this study have average TOC values (0.3 ± 0.1%) comparable to
the those of the Shihliufen Shale (0.3 ± 0.03%, n = 31), but are higher than the basal Kuantaoshan Sandstone (0.2 ±
0.1%, n = 9), and lower than the Yutengping Sandstone (0.4 ± 0.1 %, n = 216) and the Cholan Fm (0.4 ± 0.7%, n =





191; Fig. 3). C/N and $\delta^{13}C_{org}$ values of the Chinshui Shale (5.2 ± 0.7 and -24.5 ± 0.7‰, respectively) indicate stable
accumulation of marine organic content, similar to the Shihliufen Shale (5.3 ± 0.4 and -24.2 ± 0.4‰) in contrast to
the Kuantaoshan Sandstone (6.1 ± 0.3, -23.4 ± 0.3‰), Yutengping Sandstone (8.5 ± 1.8, -26.5 ± 0.5‰), as well as the
overlying Cholan Fm (6.3 ± 4.1, -25.7 ± 0.8‰), which records enhanced terrestrial input (Fig. 3). The accumulation
of marine organic matter is also stable through the Shihliufen Shale and the Chinshui shale, with greater variability
between ~4.9–4 Ma, and after ~2.3 Ma (Fig. 3).

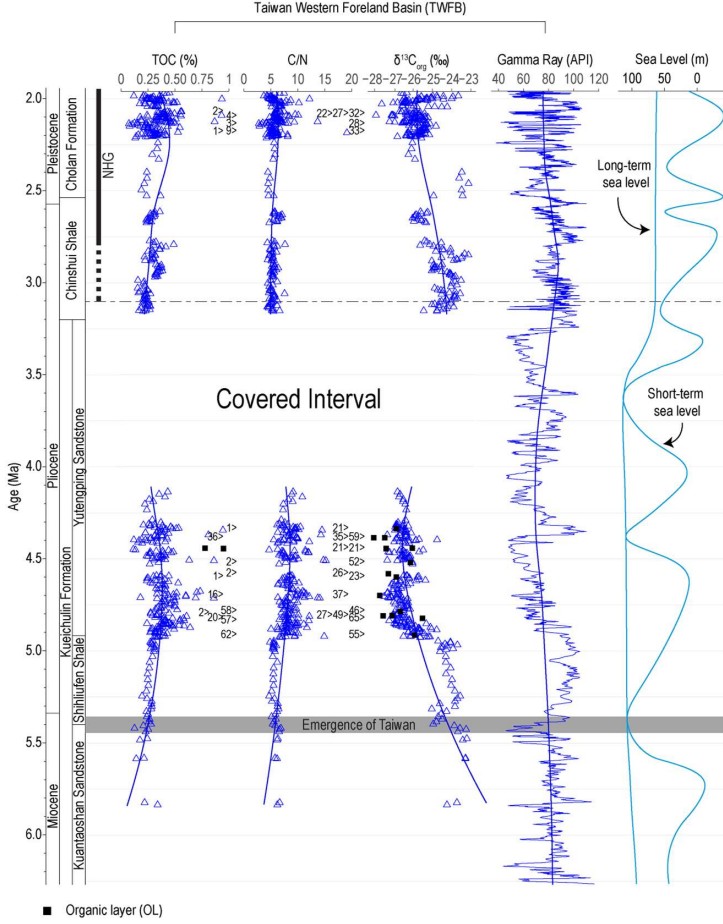


**Figure 3: Compilation of total organic carbon (TOC), C/N, $\delta^{13}C_{org}$, and gamma ray data for the Taiwan Western Foreland**
**Basin (TWFB), including the Kueichulin Fm (Dashtgard et al., 2021; Hsieh et al., 2023a; Hsieh et al., 2023b), the Chinshui**
**Shale (this study and gamma-ray from Vaucher et al. (2023b)), and the Cholan Fm (this study and gamma-ray from**
**Vaucher et al. (2023b)). Sea-level curves are from Haq and Ogg (2024). ">" indicates data that plot outside of the diagram.**
**The solid lines represent curves fitted using locally estimated scatterplot smoothing (LOESS). TOC, C/N, and $\delta^{13}C_{org}$ trends**
**reflect organic carbon sources, and show that marine organic matter content is high in the Kuantaoshan Sandstone,**
**Shihliufen Shale, and Chinshui Shale, contrasting with increased terrestrial input in the Yutengping Sandstone and Cholan**
**Formation. Gamma-ray data indicate lithological variability, and correlate with sea-level changes.**

At ODP Site 1146 (Fig. 4), MAR (n=59) and TOC (n = 225) values remain relatively stable until ~3.3 Ma (averaging
1.2 ± 0.2 g cm⁻² kyr⁻¹ and 0.08 ± 0.03%, respectively), after which both increase, with a maximum MAR of 3.5 cm⁻²
kyr$^{-1}$, and maximum TOC of 0.3%, accompanied by greater TOC variability. This is reflected in the TOC accumulation
rate (n = 225), which shows increasing trends also since ~3.3 Ma, from an average of 9.6 (± 3.7) × 10$^{-4}$ to 3.7 (± 1.8)
× 10$^{-3}$ mg cm$^{-2}$ kyr$^{-1}$. $\delta^{13}C_{org}$ (n = 113) show a gradual decrease from ~5.7–4 Ma from an average of -21.8 (± 0.4) to -
22.2 (± 0.6)‰, then stabilises. Magnetic susceptibility (n = 2747) increases through the record from an average of
~1.6 (± 0.4) to 2.5 (± 1) × 10$^{-5}$ m$^3$ kg$^{-1}$ from 5–3 Ma, with accelerated increase after ~3 Ma. Hm/Gt ratios (n = 8196)
decrease gradually from ~4.75–3 Ma (from an average of 0.56 ± 0.3 to 0.35 ± 0.1), before showing greater amplitude
variability. Gamma-ray values (n = 2551) remain relatively stable (16.2 ± 3.3 API) until ~3.2 Ma with when both
values and amplitudes rise (26.7 ± 5.7 API). The Ti/Ca record (%/%, n = 53) shows an overall decreasing trend from
~4.6 Ma–3.5 Ma from an average of 1.5 ± 0.07 to 1.2 ± 0.1.
At ODP Site 1148 (Fig. 4), MAR values (n = 15) remain stable with a slight increase at ~5.5 Ma from an average of
1.4 (± 0.009) to 1.6 (± 0.2) g cm$^{-2}$ kyr$^{-1}$, followed by a sharper increase near ~3.5 Ma to a maximum of 3.5 g cm$^{-2}$ kyr$^{-1}$
1. TOC values (n = 220), as well as TOC accumulation rates (n = 220), are stable from ~6.27–4.7 Ma (averaging 0.08
± 0.01% and 1.1 (± 0.2) × 10$^{-3}$ mg cm$^{-2}$ kyr$^{-1}$, respectively. Both TOC and TOC accumulation rates increase from
~4.7–4.5 Ma to 0.11 (± 0.01)% and 1.9 (± 0.3) × 10$^{-3}$ mg cm$^{-2}$ kyr$^{-1}$, then stabilize until ~3.5 Ma, and then increased
again (exceeding 0.2% and 5 × 10$^{-3}$ mg cm$^{-2}$ kyr$^{-1}$, respectively) with greater amplitude. MAR, TOC, and TOC
accumulation rates also exceed values measured from Site 1146 since ~4.7 Ma by 20–60%. $\delta^{13}C_{org}$ (n = 110) is broadly
stable, increasing near ~2.75 Ma from an average of -23.2 (± 0.3) to -22.8 (± 0.4)‰. Magnetic susceptibility values
(n = 1249) show a gradual increase from ~5.4–4.3 Ma from an average of 3.6 (± 0.6) to 4.9 (± 0.8) × 10$^{-5}$ m$^3$ kg$^{-1}$, then
a decrease until ~3.5 Ma to an average of 4.6 (± 1.2) × 10$^{-5}$ m$^3$ kg$^{-1}$. The values remain low after ~3.5 Ma, with
amplitudes deceasing after ~2.75 Ma. Hm/Gt (n = 1678) declines from ~5.4–4.6 Ma from an average of 0.61 (± 0.08)
to 0.2 (± 0.06), then stabilizes and slightly increases from ~3.2–2.9 Ma. Gamma-ray values (n = 1249) are high from
~5.4–4.9 Ma, averaging 29.5 (± 3.8) API, then decrease and stabilize before rising again after ~3.5 Ma to an average
of 35 (± 4.2) API. The Ti/Ca ratios (cps/cps, n = 646) increase overall from ~5.4 Ma, from an average of 0.07 (± 0.03)
to 0.16 (± 0.1), with increasing amplitude variability.

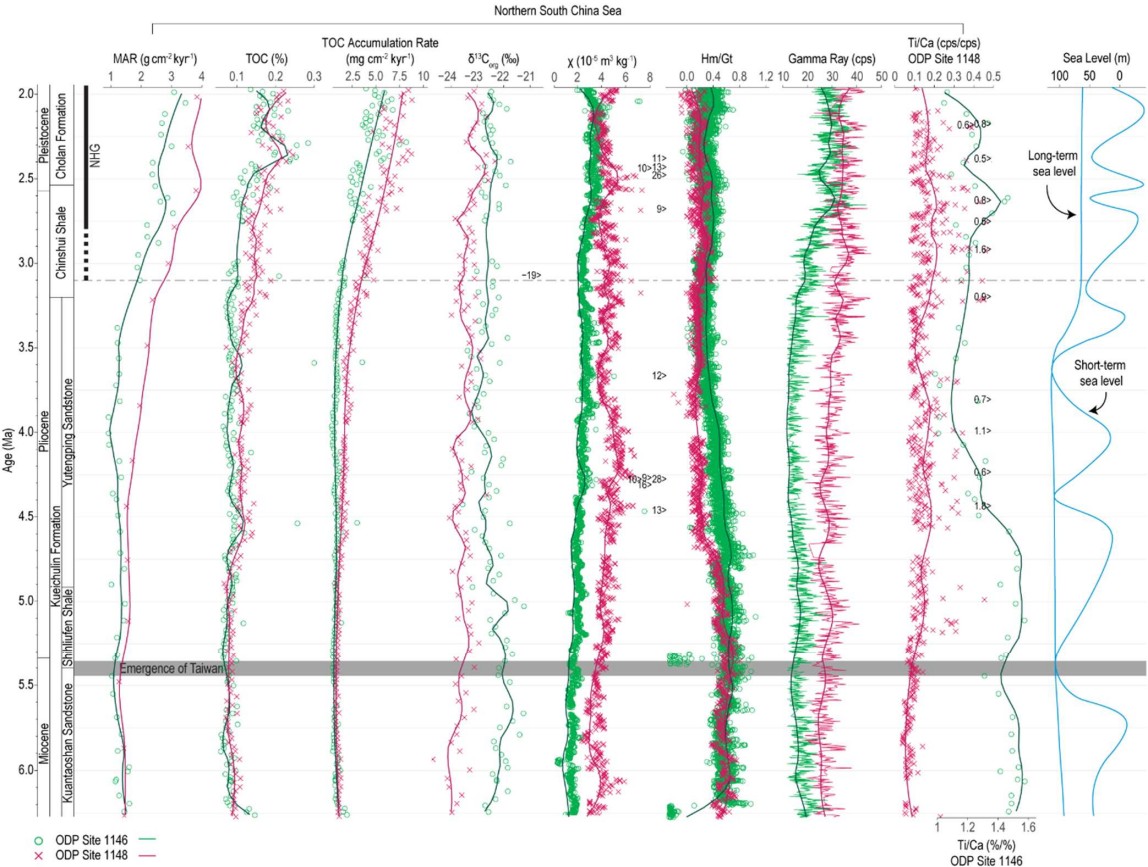

**Figure 4: Compilation of sediment core data from ODP Sites 1146 and 1148 in the northern South China Sea, including mass accumulation rate (MAR; Wan et al., 2010a; Wang et al., 2000a), TOC and δ¹³C$_{org}$ (this study), mass-specific magnetic susceptibility (χ; Wang et al., 2000a; Wang et al., 2005a), hematite/goethite (Hm/Gt; Clift, 2006; Wang et al., 2000b), gamma ray (Wang et al., 2000a, 2000b), and Ti/Ca (Hoang et al., 2010; Wan et al., 2010a). Sea-level curves are from Haq and Ogg (2024). ">" indicates data that plot outside of the diagram. The solid lines represent curves fitted using locally estimated scatterplot smoothing (LOESS). The figure illustrates the contrasting sedimentary and geochemical responses between the two ODP sites, driven by tectonic uplift, climate variability, and changes in ocean circulation.**

## 5 Discussion

### 5.1 Spatial variability in sediment provenance and distribution in the northern South China Sea

Provenance exerts a first-order control on sedimentary records in the SCS, owing to the region's complex geology and active tectonism, which channel sediment contributions from multiple major rivers (e.g., Clift et al., 2014; Clift et al., 2022; Horng and Huh, 2011; Kissel et al., 2016, 2017; Liu et al., 2009b; Liu et al., 2007; Liu et al., 2010b; Liu et al., 2016; Milliman and Syvitski, 1992; Wan et al., 2010c). During most of the Neogene, the Pearl River supplied the dominant sediment flux to the northern SCS (Clift et al., 2002; Li et al., 2003). The emergence of the Taiwan orogen in the early Pliocene fundamentally reorganised this system: by ~5.4 Ma, and especially after ~4.9 Ma, Taiwan had become a major sediment source to the adjacent TWFB and the wider SCS, as a result of rapid uplift and intense





erosion and southwestward collision-zone migration (Fig. 5; Hsieh et al., 2023b; Hu et al., 2022; Liu et al., 2010b).
This change in sediment provenance is tectonically driven and underscores the need to disentangle tectonic from
climatic signals in SCS sedimentary archives (Clift et al., 2014; Hsieh et al., 2024).
This diversity in sediment sources and mixing is reflected at ODP Sites 1146 and 1148, where the sediment records
diverge despite their spatial proximity. MAR, magnetic susceptibility, Hm/Gt and gamma-ray records diverge between
the two sites until ~3 Ma (Fig. 4). At ODP Site 1146, located on the continental slope, sediments are primarily derived
from Eurasia (Fig. 5). At Site 1146, major element and clay mineral compositions point to a mixture of sources
dominated by the Pearl River, with additional inputs from the Yangtze River, Taiwan, Luzon, and loess (Hu et al.,
2022; Liu et al., 2003; Wan et al., 2007a). Pearl River sediment discharge is controlled by long-term sea-level changes
and East Asian Monsoon variability (e.g., Liu et al., 2016), but its transport is strongly constrained: the northward-
flowing Kuroshio Current and shallow Taiwan Strait, limit delivery to the open basin, instead funnelling most material
along the continental shelf and slope via alongshore currents (Liu et al., 2010b; Liu et al., 2016; Wan et al., 2007a).

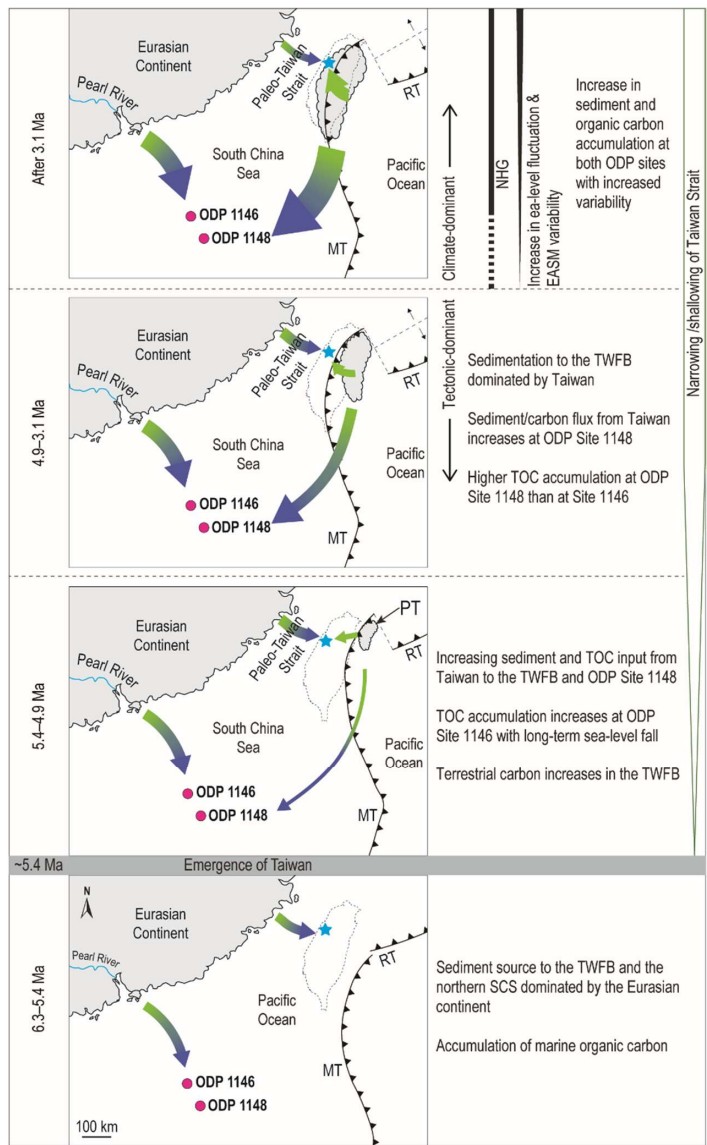


**Figure 5: Summary of different controls on sediment and carbon accumulation over time in the Taiwan Western Foreland Basin (blue star) and the ODP sites (pink circles) in the northern South China Sea. The size of the arrows indicates relative proportions of sediment flux, and green indicates accumulation of terrestrial organic carbon, while blue indicates marine organic carbon. The abbreviations MT = Manilla Trench, RT = Ryukyu Trench, and PT = proto-Taiwan. These differences in organic carbon source (i.e., terrestrial vs. marine) and carbon accumulation highlight the spatial heterogeneity in sedimentary and geochemical records within the northern South China Sea, shaped by the interplay of tectonic and climatic processes.**

In contrast, ODP Site 1148, located on the continental rise, records a stronger Taiwanese imprint (i.e., less contribution from Eurasia; Fig. 5). Prior to ~6.5 Ma, major element data suggest a mixture of Pearl River and Taiwan inputs, but since the onset of Taiwan orogenesis (~6.5 Ma), Taiwanese material has increasingly dominated (Hu et al., 2022).



Isotopic ($^{87}$Sr/$^{86}$Sr, $\varepsilon_{Nd}$), and clay mineral records corroborate Taiwan as the dominant sediment contributor to the
northern SCS since its emergence (Bertaz et al., 2024; Boulay et al., 2005; Clift et al., 2014). This conclusion is also
supported by rare-earth element studies that attribute up to 80% of slope sediments to the Taiwan orogen, and < 20%
to the Pearl River (Shao et al., 2001; Shao et al., 2009). Erosion of modern and ancient Taiwan is primarily driven by
tropical-cyclone precipitation (Chen et al., 2010; Chien and Kuo, 2011; Dashtgard et al., 2021; Galewsky et al., 2006;
Janapati et al., 2019; Vaucher et al., 2021). Under warmer Pliocene climates (Fedorov et al., 2010; Yan et al., 2016)
such storms were likely more frequent and intense (e.g., Yan et al., 2019), and especially if coinciding with EASM
circulation, would have driven exceptionally high precipitation (Chen et al., 2010; Chien and Kuo, 2011; Kao and
Milliman, 2008; Lee et al., 2015; Liu et al., 2008) and sediment export (Vaucher et al., 2023b). Sediment derived from
Taiwan is subsequently redistributed into the northern SCS by downslope deep currents (Hu et al., 2012; Liu et al.,
2013; Liu et al., 2010b; Liu et al., 2016). The emergence of Taiwan also reconfigured regional circulation, establishing
a westward Kuroshio branch that delivered additional sediment from Taiwan and the Philippines (i.e., the Luzon Arc)
into the northern basin (Liu et al., 2016).
The difference in sediment provenance and transport pathways between the continental slope and continental rise is
reflected in the contrasting proxy trends observed at both ODP sites (Fig. 4). At ODP Site 1146, the long-term increase
in magnetic minerals since ~6.27 Ma reflects increased sediment input from Eurasia that is comparatively enriched in
magnetic minerals. Concurrently, low gamma-ray values and declining Ti/Ca until ~3 Ma also reflect increased
delivery of sand-rich, clastic detritus, while the decreasing Hm/Gt suggests a progressive weakening of the EASM
rainfall and seasonality. Together, these proxy signals are consistent with global trends of long-term cooling and
falling global mean sea level during this interval (Berends et al., 2021; Haq and Ogg, 2024; Haq et al., 1987; Holbourn
et al., 2021; Jakob et al., 2020; Miller et al., 2020; Rohling et al., 2014; Wan et al., 2007b; Westerhold et al., 2020),
as well as with evidence of diminished chemical weathering and progressive weakening of the EASM system (Clift,
2025; Clift et al., 2014; Li et al., 2004; Wan et al., 2006; Wan et al., 2010a; Wan et al., 2010b; Wang et al., 2019).
This interpretation is further supported by declining K/Al ratios observed at ODP Site 1146 between 5 and 3.8 Ma by
Tian et al. (2011), which likewise indicate reduced chemical weathering and a shift towards long-term drying.
At ODP Site 1148, MAR increases near the onset of Taiwan's orogenesis (~5.4 Ma), reflecting enhanced sediment
export from rapid erosion the emerging orogen. An increase in magnetic susceptibility is also observed ~5.4–4.3 Ma
(Fig. 4), consistent with the erosion of passive-margin seafloor sediments enriched in magnetic minerals that was
uplifted during the early stages of Taiwan's orogenesis (Hsieh et al., 2023b). After ~4.3 Ma, magnetic susceptibility
declines, coinciding with the deposition of the Yutengping Sandstone and increasing influx of sediment derived from
the metasedimentary core of Taiwan, which is comparatively depleted in magnetic minerals (Hsieh et al., 2023b).
Unlike Site 1146, the Hm/Gt record at Site 1148 does not appear to track long-term the monsoon drying. Rather, the
abrupt decrease in the Hm/Gt record at ~5.4 Ma is attributed to the influx of hematite-depleted sediment from Taiwan
as it emerged from the Pacific Ocean. The dispersal of Taiwan-sourced sediment into the northern SCS was facilitated
by deep-water currents and by the westward-flowing Kuroshio Branch, both of which developed following the
formation of the Taiwan and Luzon straits during orogenesis. Changes in ocean circulation during the early to middle
Pliocene are also captured by K/Al records, which show contrasting trends between intermediate water depths (e.g.,



Site 1146) and deep water settings (e.g., Site 1148), which is interpreted as reflecting shifts in sediment dispersal
pathways to the northern SCS (Tian et al., 2011). The subsequent rise in Hm/Gt near ~3.2 Ma is attributed to the
northward remobilization of Taiwan-sourced sediment following the formation of Taiwan Warm Current (Fig. 3;
Hsieh et al., 2024). The gamma-ray record also tracks the orogenic evolution of Taiwan at both ODP sites (Fig. 4) and
parallels observations from the TWFB (Fig. 3): values are elevated during the deposition of mudstone-rich Shihliufen
Shale, decrease during formation of sand-dominated Yutengping Sandstone and rise again with the deposition of
mudstone-rich Chinshui Shale and Cholan Fm. The increase in sediment export from Taiwan is also reflected in the
Ti/Ca record, which increases after ~5.4 Ma, in response to intensified physical erosion and elevated terrestrial flux
linked to the onset of Taiwan orogenesis.
After ~3 Ma, the onset of Northern Hemisphere Glaciation (NHG) resulted in enhanced seasonality and an
intensification of the EASM (Fig. 5; Clift, 2025; Clift et al., 2014; Wan et al., 2006; Wan et al., 2007a; Wan et al.,
2007b). Although global cooling characterized the late Plio-Pleistocene (Lisiecki and Raymo, 2005), sea-surface
temperatures in the northwest Pacific remained sufficiently high (26.5–27.0°C) to sustain tropical cyclone activity
(Tory and Frank, 2010). This combined influence of intensified EASM and frequent tropical-cyclone precipitation
promoted elevated sediment production and large-scale export of fine-grained material enriched TOC from river
catchments into offshore depocenters. This is reflected in both sites by higher gamma-ray values, increased MAR, and
rising Ti/Ca ratios (Fig. 4). Enhanced seasonality is further expressed in the greater amplitude observed in gamma-
ray, Hm/Gt, and Ti/Ca records.
**5.2 Influence of terrestrial sediment export vs. primary production on carbon burial**
Organic carbon buried in the SCS can be broadly divided into two components: (1) terrestrial organic matter derived
from rock, soil, and terrestrial vegetation exported from adjacent landmasses by precipitation-driven erosion, and (2)
marine organic matter produced by primary productivity and exported to the seafloor.
At Site 1146, organic carbon accumulation, like bulk sediment accumulation, is primarily controlled by long-term
global sea-level fall associated with the onset and intensification of NHG (Fig. 5). Total organic carbon values are
closely coupled with MAR, with increases in sediment flux consistently accompanied by higher TOC concentrations
(Fig. 4). Although $\delta^{13}C_{org}$ values show a modest decline between ~5.7 and 4.5 Ma, which is consistent with episodic
dilution by terrestrial organic inputs, values remain within the marine range (Table 1). The gradual increase in
terrestrial organic matter at ODP Site 1146 is interpreted to reflect increased Eurasian clastic influx under conditions
of long-term sea-level fall. The cross-plot of $\delta^{13}C_{org}$ and TOC also shows no distinct shift between organic matter
delivered to Site 1146 before and after the emergence of Taiwan. As sediment transported eastward from the Eurasian
margin would have longer residence times in the ocean, the dilution of land-derived organic material by marine organic
material would increase, resulting in a more marine $\delta^{13}C_{org}$ signature (Dashtgard et al., 2021) which supports the
interpretation that organic material is derived mainly from Eurasia via the Pearl River (Fig. 6).



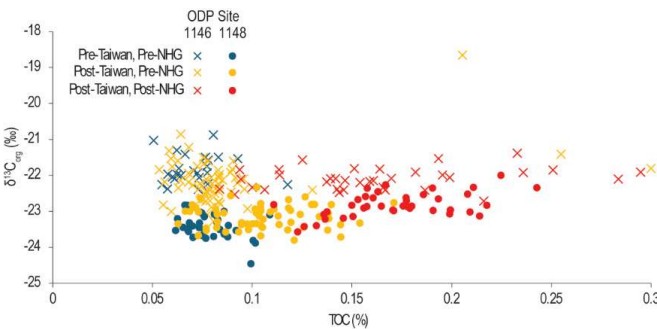


**Figure 6: Cross-plot of $\delta^{13}C_{org}$ and TOC measured from ODP Sites 1146 and 1148. Values are grouped according to major tectonic and climate changes: 1) pre-emergence of Taiwan and pre-Northern Hemisphere Glaciation, 2) post-emergence of Taiwan and pre-Northern Hemisphere Glaciation, and 3) post-emergence of Taiwan and post-Northern Hemisphere Glaciation. Note the distinct trends before and after Taiwan's emergence and Northern Hemisphere Glaciation. Site 1146 reflects Eurasian sediment input with marine organic matter dominance, while Site 1148 highlights Taiwan's influence, with enhanced marine productivity linked to nutrient export.**

In contrast, carbon burial at Site 1148 is primarily linked to the uplift and erosion of Taiwan and associated increase in sediment and nutrient delivery to the marine environment (Fig. 5). The onset of orogenesis in Taiwan at ~5.5 Ma coincides with a marked rise in MAR, followed by an increase in TOC beginning near ~4.9 Ma (Fig. 4). This pattern indicates significant export of terrestrial sediment from the rapidly uplifting Taiwan orogen, a process further amplified by the coupling between tropical cyclone and monsoon precipitation (Vaucher et al., 2023b). Notably, TOC increases proportionally with MAR, implying that carbon burial was not diluted by high sediment flux but rather enhanced by intensified sediment export, highlighting the role of Taiwan as a contributor of organic carbon in the northern SCS. The influence of sedimentation from Taiwan on organic matter buried at Site 1148 is also evident from the cross-plot between $\delta^{13}C_{org}$ and TOC, which shows a distinct increase in TOC prior to and after the emergence of Taiwan (Fig. 6).

Taiwan's steep topography and active tectonics generate exceptionally high sediment yields to adjacent marine systems (Dadson et al., 2004; Dadson et al., 2003; Liu et al., 2013). Turbidity currents, especially via submarine canyon systems (e.g., the Gaoping Submarine Canyon in southern Taiwan), efficiently transport organic-rich sediment eroded from Taiwan to deep-sea environments approximately 260 km offshore into the northeastern Manila Trench (Liu et al., 2009a; Liu et al., 2016; Nagel et al., 2018; Yu et al., 2009; Zheng et al., 2017). Within the TWFB, this process is manifested as an abrupt increase in terrestrial organic matter and sand-rich deposition near ~4.9 Ma with the emplacement of the Yutengping Sandstone (Fig. 4). At Site 1148, TOC increases markedly in association with the emergence of Taiwan, and $\delta^{13}C_{org}$ values remains stable above -25‰. While $C_4$ plants are characterized by high $\delta^{13}C_{org}$ values (Table 1), and an expansion of $C_4$ plants in the South China region has been documented since 35 Ma (Li et al., 2023; Xue et al., 2024), the organic carbon at Site 1148 are interpreted to be of marine in origin as $C_3$ plants remain the dominant vegetation type in the study area (Luo et al., 2024; Still et al., 2003; Wang and Ma, 2016). Furthermore, sediment provenance markers (Section 5.1) indicate an influx of Taiwan-sourced material to Site 1148 after the emergence of Taiwan, and $\delta^{13}C_{org}$ values in the TWFB reflect an increase in terrestrial organic matter. The presence of Taiwan-sourced material combined with high proportions of marine organic carbon at Site 1148 suggests that terrestrial organic matter from Taiwan was largely confined to proximal coastal environments, and that enhanced





carbon burial in deeper settings reflects processes beyond direct terrigenous input. Likewise, terrestrial organic matter
contribution from the Pearl River into deeper-water depocenters is limited, as sediment is dispersed along the
continental shelf by alongshore currents (Liu et al., 2010b; Liu et al., 2016; Wan et al., 2007a). During transport and
sedimentation, degradation does not appear to significantly alter the isotopic composition of organic matter, since
there is little fractionation between reactants and products. If post-depositional alteration were a dominant control,
$\delta^{13}C_{org}$ values should become progressively less negative with depth, as lighter isotopes are preferentially removed.
However, the $\delta^{13}C_{org}$ records from the two sites show distinct trends, suggesting that the influence of post-depositional
isotopic fractionation is insignificant.
Taiwan's rapid denudation delivers large quantities of sediment and nutrients to the northern SCS, profoundly shaping
basin productivity and carbon cycling. The export of bioessential nutrients stimulates intense coastal primary
production, as reflected by modern chlorophyll-a and nitrogen distributions that peak along Taiwan's coast before
rapidly declining offshore due to swift uptake (Ge et al., 2020; Huang et al., 2020; Kao et al., 2006). Episodic inputs
from tropical cyclones, which contribute up to 80% of summer particulate organic carbon, further amplify productivity
and promote lateral dispersal of sediments (Liu et al., 2013). Marine organic matter produced through enhanced coastal
productivity could be redistributed by deep-water contour currents and mesoscale eddies, (Hsieh et al., 2024; Lüdmann
et al., 2005; Zhang et al., 2014; Zhao et al., 2015), enabling its bypass into the deeper water depths and resulting in
the marine signature of the $\delta^{13}C_{org}$ records from the northern SCS,.
Fluvial input from Taiwan, especially via submarine canyon systems, makes the northern SCS a depocenter for organic
carbon burial, with important implications for the basin's sedimentary architecture, long-term carbon budget, and even
hydrocarbon source rock potential (Kao et al., 2006). Paleoceanographic records indicate that productivity and organic
carbon burial increased during glacial periods (Thunell et al., 1992), likely driven by nutrient delivery from Taiwan's
sediments that enhanced the biological pump and contributed to regional carbon drawdown. In the modern setting,
episodic sediment fluxes during typhoons sustain unusually high chlorophyll-a concentrations in deep SCS waters
relative to the global ocean (Shih et al., 2019). Moreover, northeast monsoon-driven mixing between the China Coastal
Current and Taiwan Strait Current, reinforced by sediment and nutrient inputs from Taiwan and the Yangtze River,
sustains elevated productivity in the northern SCS (Huang et al., 2020). Collectively, these processes highlight
Taiwan's sediment flux as a key linkage between monsoon forcing, nutrient cycling, and primary production across
both modern and in the past.
**5.3 Influence of climate and monsoon on carbon burial**
In the TWFB, carbon geochemistry and gamma-ray data largely reflect the evolution of the foreland basin
synchronously with the shifts in the regional climate regime (Fig. 3). During the deposition of the Chinshui Shale in
the late Pliocene (~3.2 to 2.5 Ma), reconstructions for the northwest Pacific show relatively high global sea levels and
stable sea-surface temperatures (Berends et al., 2021; Li et al., 2011). Such conditions favoured the accumulation of
fine-grained sediment, while elevated sea levels deepened the TWFB and promoted offshore depositional
environments-both of which are expressed in the Chinshui Shale (e.g., Nagel et al., 2013; Vaucher et al., 2023b.
Greater water depths and increased distance from the terrestrial sediment sources also enhanced the relative





contribution of marine organic matter. The gamma-ray record of the TWFB strata further reveals depositional cycles
related to interactions between EASM and tropical cyclone precipitation after ~4.92 Ma, with variability expressed at
both short-eccentricity and precession frequency bands (Hsieh et al., 2023a; Vaucher et al., 2023b).
During the early Pleistocene, with deposition of the Cholan Fm (~2.5–1.95 Ma), global sea level and regional sea-
surface temperatures became markedly more variable (Berends et al., 2021; Li et al., 2011). The continued uplift and
southwest migration of Taiwan promoted the development of shallow-marine depositional environments recorded in
the Cholan Fm (e.g., Pan et al., 2015; Vaucher et al., 2023a; Vaucher et al., 2023b; Vaucher et al., 2021). This is
expressed in the gamma-ray and carbon records as an increase in terrestrially sourced, sandstone-rich intervals with
high variability (Fig. 3). The enhanced in export of coarser-grained sediment from land to sea is likely related to the
onset of NHG, when repeated sea-level minima promoted clastic delivery to the basin (Vaucher et al., 2023b; Vaucher
et al., 2021). In addition, global climate deterioration related to NHG intensified and destabilised the EASM, which
would in turn increase sediment supply to the South China Sea (Wan et al., 2006; Wan et al., 2007a).
In the northern SCS, MAR and TOC values and amplitudes at both ODP sites increased after ~3 Ma, consistent with
increased sediment export (Fig. 4; Fig. 5). Paleoclimate reconstructions from East Asia likewise document a
strengthening of the EASM during the late Pliocene, generally near ~3.5 Ma (Hoang et al., 2010; Nie et al., 2014; Xin
et al., 2020; Yan et al., 2018; Yang et al., 2018; Zhang et al., 2009). While the causal relationship between monsoon
intensification and NHG remains debated (Nie et al., 2014; Wan et al., 2010b; Xin et al., 2020; Zhang et al., 2009),
long-term global cooling and sea-level fall coupled with intensified monsoon and tropical cyclone precipitation likely
acted together to amplify sediment export from land to sea (Vaucher et al., 2023b). At the same time, $\delta^{13}C_{org}$ values
at ODP Site 1148 increases after ~3 Ma, suggesting increasing marine contribution to organic carbon. This trend is
attributed to enhanced marine primary production driven by nutrient enrichment. Independent evidence for increased
marine primary productivity in this interval comes from elevated abundances of planktonic foraminifera
*Neogloboquadrina dutertrei* and higher biogenic silica production (Wang et al., 2005b).
**6 Conclusion**
Analyses of late-Miocene to early Pleistocene sedimentary and geochemical records from shallow-marine strata of
the Taiwan Western Foreland Basin and deep-sea sediment cores from the northern South China Sea (SCS) provide
clear evidence for shifting pathways of carbon erosion, transport, and burial shaped by the interplay between tectonic
forcing, climate variability, and oceanographic processes.
Sediment provenance reveals marked spatial heterogeneity between the continental slope (ODP Site 1146) and the
continental rise (ODP Site 1148), highlighting the influence of tectonic uplift and evolving ocean circulation on
sediment mixing and deposition. Prior to ~5.4 Ma, sediment delivery to the northern SCS was dominated by Pearl
River discharge. Taiwan's rapid emergence and erosion at ~5.4 Ma supplied large volumes of clastic material to the
basin, which is expressed in sediment provenance records at Site 1148, whereas Site 1146 remained strongly
influenced by Eurasian sources. Pearl River sediments were dispersed along the continental shelf and slope by
alongshore currents but were largely obstructed from reaching deeper water depths by the northward-flowing Kuroshio
Current and the shallow Taiwan Strait.



The onset of Northern Hemisphere Glaciation (NHG; ~3 Ma) further amplified sediment erosion and export across
the basin. Long-term global cooling and sea-level fall, coupled with enhanced seasonality, drove the intensification of
the East Asian Summer Monsoon. The resulting increase in monsoon rainfall, as well as persistent tropical cyclone
activity, drove synchronous increases in mass-accumulation rate (MAR), magnetic susceptibility, and Ti/Ca values at
both ODP sites, demonstrating the strong climatic imprint on sediment export. In addition, slightly higher $\delta^{13}C_{org}$
values after ~3 Ma indicate a greater marine contribution to organic matter, attributed to enhanced nutrient-driven
marine primary production.
Organic carbon burial likewise reflects the combined influence of tectonic and climate forcing. At ODP Site 1146,
total organic carbon (TOC) accumulation parallels MAR and is primarily controlled by long-term sea-level fall and
NHG intensification. $\delta^{13}C_{org}$ values indicate that the bulk of organic matter remained marine in origin, with minor
terrestrial contribution linked to Eurasian sediment export rather than to local tectonics. At ODP Site 1148, by contrast,
organic carbon burial is closely tied to the Taiwan's uplift and erosion. Importantly, TOC scales proportionally with
MAR, implying that organic matter burial was enhanced—not diluted—by high sediment flux. Despite Taiwan's steep
relief, rapid tectonic uplift, and frequent typhoon- and monsoon-driven erosion generating exceptional sediment
yields, $\delta^{13}C_{org}$ values indicate that most buried organic was marine. This suggests that Taiwan's erosion enhanced
nutrient supply, stimulating coastal primary productivity. Marine organic matter produced in these settings was then
redistributed offshore by turbidity currents through submarine canyon systems, bypassing the shelf and slope and
accumulating in deep-sea depocenters of the northern SCS.
Overall, this study highlights the importance of resolving spatial heterogeneities in sedimentary climate archives.
Disentangling the competing influences of tectonic and climate on sediment supply and carbon burial is critical for
robust intercomparison of paleoclimate records, and for reconciling apparent inconsistencies among proxy
reconstructions. Our findings also demonstrate that terrestrial sediment export contributes to carbon drawdown via
two distinct pathways: (1) direct burial of eroded terrestrial organic matter and (2) nutrient supply that fuels marine
primary production and subsequent burial of marine organic matter. This work establishes a direct link between the
tectonic evolution of an arc-continent collisional orogen and changes in carbon storage in adjacent basins, and
disentangles the mechanisms by which the erosion of mid-latitude orogens contributed to long-term carbon
sequestration.
**Data availability**
The data that support the findings of this study will be submitted to PANGAEA upon acceptance.
**Author contribution**
A.I.H. was responsible for the design and conceptualization of this study, supervised by S.J. Data collection was
completed by A.I.H., S.B., and R.V. A.I.H., T.A., L.L., B.B., L.K., and P.-L.W. were responsible for sample analysis.
T.A., L.L., S.B., R.V., and S.J provided support in the interpretation of sedimentary paleoenvironmental proxies. All
co-authors reviewed and approved the manuscript.



**Competing interests**
The authors declare that they have no conflict of interest.
**Acknowledgements**
We would like to thank Dr. Yusuke Kubo at the Japan Agency for Marine-Earth Science and Technology for access
to the ODP Site 1146 and 1148 core samples. We express our gratitude to Kuo-Hang Chen for supporting the
magnetostratigraphic analysis, Tiffany Monnier for assisting in sample processing and analysis, and Ling-Wen Liu
for elemental and isotope analyses. We are grateful for the constructive feedback from ________, as well as the
support from the editor, ________ who helped to greatly improve this manuscript.
**Financial support**
This research was supported financially through the Institute of Earth Sciences Postdoctoral Fellowship awarded to
A.I. Hsieh from the University of Lausanne. R. Vaucher acknowledges the Swiss National Science Foundation
Postdoc.Mobility Grant (P400P2_183946) which supported him during data collection from the Cholan Fm.

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
