# Peer review of "Carbon export and burial pathways driven by a low-latitude arc- 2 continent collision"

_EGUsphere, 2025_

## Author Comment (AC1)

**University of Lausanne**
**Institute of Earth Sciences**
Unil-Mouline, Géopolis
Route de la Mouline 11
1015 Lausanne

Dear Prof. Soreghan and reviewers,

Please find below, our responses to reviewer comments for the following manuscript, in blue:

Carbon export and burial pathways driven by a low-latitude arc-continent collision

Amy I. Hsieh, Thierry Adatte, Shraddha Band, Li Lo, Romain Vaucher, Brahimsamba Bomou, Laszlo Kocsis, Pei-Ling Wang, Samuel Jaccard
* * *
Reviewer #1

This is an intriguing study that looks at the impact of an arc-continent collision on regional carbon budgets and particularly the export of terrestrial organic material from continental southern China and from Taiwan. Taiwan is the world type example of arc-continent collision. The study compares the flux of organic carbon from the land with the amount of organic material of marine origin which is buried in the deep South China Sea. The paper reaches an interesting conclusion that much of the buried organic material is of marine origin and that greater rates of carbon burial are achieved as a result of the delivery of nutrients from the land into the coastal ocean increasing productivity which is then buried by the significant flux of clastic sediment, especially from Taiwan. In general the work is useful in noting the potential importance of arc-continent collision zones as being places with significant amounts of organic carbon may be sequestered although any material buried like this in the South China Sea is liable to be recycled in the relatively near geological future because of the Southeast migration of Taiwan through time, meaning that the storage is not long lived. I was only partly convinced that two neighbouring drill sites really had quite such distinct provenance with one being almost exclusively related to sediment delivered by the Pearl River and the other from Taiwan, but I think even if we take this with a pinch of salt that the overall conclusions are interesting in showing the coupling between the tectonics and the burial of organic carbon and the potential influence of high amplitude climatic variations modulated through the Asian monsoon and linked to Northern hemisphere glaciation. There was no estimate of the total flux it so it's not possible to see whether this region might have a global impact although it seems unlikely based on recent work. I had a number of smaller questions which I left below and which the authors are encouraged to respond to.

1. Line 13 - In low-latitude regions, monsoon - Monsoon is more of a mid latitude phenomenon

   Edited to "mid- to low-latitude regions"

2. Line 15 - physical weathering - You mean erosion?

   Edited to "physical erosion"

3. Line 18 - accumulation controlled by long-term sea-level fall and glaciation.  – You mean because sediment is stored on the shelf during sea level highstands? Is Taiwan glaciated?

   Changed to "long-term sea-level fall and shoreline progradation"

4. Line 22 - Marine organic matter along Taiwan's shore was subsequently remobilized - How do you know the organic matter is not produced further from the coast and just falls out through the water column to be preserved in deeper water?

   This interpretation is based on modern chlorophyll-a and nitrogen distributions that show peaks along Taiwan's coast but rapidly decrease offshore (Section 5.2). This suggests that marine organic matter would have been produced near Taiwan's shoreline rather than farther from the coast.

5. Line 24 - intensification of the East Asian Summer Monsoon - Yes but only during interglacial times

   Edited to specify that the East Asian Summer Monsoon intensified during interglacial periods

6. Line 29 - erosion of an arc-continent collision - Yes but anywhere with fast sedimentation has the potential to busy more organic material, See Pleistocene of the Bengal Fan.

   Clift, P.D., Jonell, T.N., Du, Y., Bornholdt, T., 2024. The impact of Himalayan-Tibetan erosion on silicate weathering and organic carbon burial. Chem. Geol., 656: 122106. doi:10.1016/j.chemgeo.2024.122106.

   Agreed, although high sedimentation rates can also dilute organic matter concentrations. The chemical weather of arc-continent collisional orogens has been hypothesized to have regional/global influence on the carbon cycle, yet the relationship between the two have only been established by indirect comparison of proxy-based records/reconstructions. While a positive relationship between erosion and organic carbon burial has been established in regions with rapid sedimentation rates (e.g., Bengal Fan), given the regional/global implication, we believe it's crucial to establish the link specifically between an active arc-continent and carbon burial.

7. Line 34 - during the Neogene (Clift and Jonell, 2021) - Yes but also see more recent work by same authors. Clift et al. (2024)

   Included Clift et al. (2024) in the cited literature

8. Line 47 - that were considerably more intense and frequent than at present - But we are in an interglacial. Why would typhoons be stronger during previous interglacials?

   For clarity, the sentence now states, "Warm sea-surface temperatures and reduced polar ice volumes under greenhouse would likely amplify monsoon variability and produce frequent and intense tropical cyclones."

9. Line 70 - Since its emergence in the early Pliocene, Taiwan - Emergence in this location. The collision zone has been migrating SW through time so there would have been an emergent island before then but located further to the NE.

   We agree that the collision zone would have been propagating towards the southwest prior to this time, but geochemical records in the Taiwan Western Foreland Basin (Hsieh et al., 2023 Basin Research), which record the early stages of Taiwan's orogenesis and emergence from the Pacific Ocean, suggest that while Taiwan's orogenesis (i.e., the migration of the collision zone to near it's present-day location) began in the late-Miocene, the island was not fully emergent until the early Pliocene.

10. Line 86 - late Pliocene = upper Pliocene

    For consistency, we edited the geologic ages with respect to the time interval using "early" and "late" instead of "lower" and "upper". The text has been edited to reflect this (e.g., line 80, "The base of the TWFB stratigraphic fill is composed of the Kueichulin Formation (Fm), a sandstone-dominated unit *deposited between the late Miocene–early Pliocene*…").

11. Line 92 - The targeted time interval (~6.27–1.95 Ma) – Why was this time chosen?

    We have changed the wording of this paragraph to reflect our rationale for targeting this time interval.

    *The time interval between ~6.27–1.95 Ma was targeted because it spans the initiation of Eurasian-Philippine plate collision through the mergence and uplift of Taiwan. It also includes the Pliocene (5.33–2.58 Ma), which may be the most recent time in Earth's history when atmospheric $CO_2$ last reached or exceeded present-day concentrations (>400 ppm; Tierney et al., 2019), and the subsequent transition toward Pleistocene icehouse conditions. Additionally, since tectonic configurations, insolation, and major floral and faunal assemblages have remained broadly unchanged since the mid-Pliocene (Dowsett, 2007; Robinson et al., 2008), this period also provides a critical Earth system analogue for evaluating future climate hazards (e.g., Burke et al., 2018), including sea-level rise and extreme weather events.*

[Figure]

12. Line 92 - spans the initiation of Eurasian-Philippine plate collision – No. Only in this place Its not the first time of collision

    We have edited the sentence to be more precise

    …it spans the initiation of Taiwan's emergence and uplift by the ongoing Eurasian-Philippine plate collision.

    This specifies that the time period covers Taiwan's emergence from the Pacific Ocean but clarifies that the collision between the Eurasian Plate and the Philippine Sea Plate was initiated prior to this time.

13. Figure 2 - Dark writing against dark background is hard to read.  Why not have a blue shaded bathymetry?

    We changed the font and lightened the background so that the text should be easier to read.

14. Line 113 - Da'an River – If you are going to name geographic features you need to show them on a map. Likewise, Tachia River, Houlong River

    The locations of the outcrops are shown in Figure 2A in the inset map.

15. Line 116 – ODP - Explain abbreviation.

    The abbreviation is defined in section 1 Introduction.

16. Line 120 - ODP Sites 1146 and 1148 cores – Why do you think these contain Taiwan material and not just from southern China?

    While both ODP Sites 1146 and 1148 likely received sediment from multiple sources, sediment provenance indicators from ODP Site 1148 show that Taiwan became a major source of sediment relative to Eurasia as it emerged from the Pacific Ocean (Hsieh et al. 2024, Palaeogeography, Palaeoclimatology, Palaeoecology). In contrast, at ODP Site 1146, the provenance indicators suggest that the major source of sediment is Eurasia, and that the influence of Taiwan's emergence on the sediment source is negligible (this study).

17. Line 131 - achieving a precision of better than 0.3% (REFS) - REFS??

    We have removed this typo.

18. Line 155 - Thirty-three oriented palaeomagnetic core specimens (25-mm diameter) were collected at ~3.5 m intervals – But from where?

    We clarified that the samples were collected from the Chinshui Shale.

19. Line 165 - serve as proxies for physical erosion - this is not justified. The MAR at a single site cannot be considered proxy for the amount of physically erosion in the source areas. There are lots of factors not least sediment transport in the deep water and accommodation space issues on the continental shelf that mean that these are undoubtedly unrepresentative.

    We have removed MAR as a proxy for physical erosion. Though at both ODP Sites, MAR increases proportionally with physical erosion, suggesting that locally at least, physical erosion is the dominant control on MAR.

20. Line 170 - sourced from the South China – you need to be more specific. Where is the South China data from? If you use material from the Pearl River then that is not acceptable because the river is highly anthropogenically disrupted and not representative of the flux in the past.

    We clarified that the sediment sources are along the Taiwan Strait region (i.e., north of the Pearl River).

[Figure]

21. Line 178 - Sedimentary TOC content provides a measure of organic carbon accumulation through time – yes, but as for the clastic mass accumulation it is not acceptable to use the rate of organic carbon MAR at a single site as approximately for the total flux

    Sedimentary TOC can provide insights into organic carbon deposition through time, however, it alone cannot be used to determine the amount of carbon accumulated in the sedimentary record though time. Carbon mass accumulation rates were calculated to evaluate if increased sediment flux served to dilute or concentrate carbon, which we believe is a more robust indicator of carbon accumulation.

22. Line 189 - Hematite typically forms through iron oxidation under arid climates – that's not completely correct. It may be indicative of a seasonal climate with a dry season.

    Corrected to include reviewer's feedback.

    *Hematite typically forms through iron oxidation under arid climates or seasonal climates with dry seasons…*

23. Line 218 - Stratal ages from ODP Site 1148 (Clift, 2006) are constrained using biostratigraphic ages of benthic foraminifera (Wang et al., 2000a) – the age model is from Wang et al.

    We have corrected the sentence according to the reviewer's feedback.

    *Stratal ages from ODP Site 1148 are constrained using biostratigraphic ages of benthic foraminifera (Wang et al., 2000).*

24. Line 221 the those - Delete "the"

    We deleted the redundant "the"

25. Line 227 - Chinshui shale - Chinshui Shale

    We reviewed Line 227 and confirm that the correct intervals should be the Shihliufen Shale and the Chinshui Shale.

26. Figure 3 – why is the label "covered interval" so enormous compared to the rest of the text?

    We have edited the font size of Figure 3 for consistency.

27. Line 239 - after which both increase, with a maximum MAR of 3.5 cm—that only tells you that the accumulation increased in this particular location but it doesn't tell you anything about the discharge from the river. Sedimentation rates at a single location being governed by a variety of factors. For example the increase may simply refer to moderate change in the peed and location of a contour current.

    We agree with the reviewer that multiple factors can influence sedimentation rates, however, this is a part of the results section which only describes the trends of the different parameters shown in Figure 4. While we agree that changes to MAR are not attributed solely to changes in fluvial discharge, the timing of a major increase in MAR at Site 1148 coincides with the emergence of Taiwan (and continues to increase thereafter), and the timing of the accelerated increases in MAR at both ODP Sites after ~3 Ma coincide with the onset of Northern Hemisphere Glaciation. The increases in MAR also correspond to indicators of physical erosion (e.g., magnetic susceptibility, Ti/Ca), which suggest that there is a local relationship between MAR and terrestrial sediment export. As well, the emergence of Taiwan also resulted in the formation of and strengthening of deep-water currents, which served to distribute sediment farther into the South China Sea.

28. Line 275 - which channel sediment - which channels sediment

    Corrected to "channels"

[Figure]

29. Lines 285 - sediment records diverge despite their spatial proximity - I'm not sure I agree. The records parallel each other. They are not identical but there are plenty of local reasons that MAR would not be identical and even TOC could be affected by local dilution by clastic or biogenic sedimentation.

    We changed "diverge" to "differ". As the reviewer said, the records are not identical due to various reasons, of which we highlight changing sediment provenance.

30. Line 290 - Pearl River sediment discharge is controlled by long-term sea-level changes and East Asian Monsoon variability (e.g., Liu et al., 2016) - Sea level does not control the discharge. Only where the sediment is deposited. The monsoon impact on Pearl River discharge was noted long before Liu et al. (2016), e.g., Clift (2006)

    We changed "discharge" to "deposition" and added the reference to Clift (2006).

31. Line 291 - shallow Taiwan Strait – What does the Taiwan Strait have to do with the Pearl River? The modern/Holocene flux is to the west. As noted by

    Liu, J. P., Z. Xue, K. Ross, H. Wang, J., Z. S. Yang, A. C. Li, et al. (2009), Fate of sediments delivered to the sea by Asian large rivers: Long-distance transport and formation of remote alongshore clinothems, Sed. Record, 7(4), 4-9.

    At the LGM flux was to the deep SCS via the Canyon.

    We have corrected the structure of this statement for clarity.

    Liu et al. (2009) shows a conceptual figure of sediment flux from the Eurasian margin and Taiwan westwards to the sea, but detailed reconstructions (e.g., Dadson et al. 2003, Milliman et Kao 2005, Kao et al. 2008, Liu et al. 2016) show large volumes of sediment transported from Taiwan towards the south and west. The isopach map from Liu et al. (2009) and their interpretation that "the majority of Pearl River sediments are trapped inside the estuary, although sediments that do escape to the shelf are transported alongshore but have not yet formed a large remote nearshore depocenter" also agrees with our findings.

32. Figure 5 - Why do you show Taiwan getting smaller in the past? The collision is migrating to the SW but there is no reason to believe the collisional orogen was smaller in the past

    Please refer to comment #9. In the earliest stages of Taiwan's emergence, the surface area and elevation should have been smaller than it is today. Thus, our figure represents a conceptual model of Taiwan's growth and does not reflect its actual size in the geologic past.

33. Line 302 - major element data suggest a mixture - Major element data is not a good provenance tool. Its mostly affected by weathering. Also, isotopic (87Sr/86Sr, εNd) methods are not suitable either because the sources are all from South China and Sr is affected by weathering which changes with climate.

    We agree that major element, [87]Sr/[86]Sr, εNd, and indeed, all proxy records, are susceptible to alteration as a result of weathering. We account for the influence of weathering on the records by adopting a multi-proxy approach. Given that many proxies at ODP Site 1148 broadly align with the timing of Taiwan's emergence, we believe that the change in sediment provenance is a major control and is manifested in these proxies.

34. Line 304 - clay mineral records - Specifically what do you mean?

    We refer to the increase in illite and corresponding decrease in kaolinite near 5 Ma (now stated in the manuscript).

35. Line 306 - supported by rare-earth element studies – these are not reliable because of the common sources to these areas and using modern Pearl River sediment is a bad fingerprint for the past because of anthropogenic disruption of the catchment.

[Figure]

We have removed references to these studies of modern sedimentation as they may not be relevant to studies of the geologic past, as the reviewer recommended.

36. Line 319 - Increase in magnetic minerals since ~6.27 Ma reflects increased sediment input from Eurasia - How do you know this is a sign of Eurasian input?

We refer to the study by Horng and Huh (2011) and Hsieh et al. (2023) which show an enrichment in magnetic minerals in Eurasia-sourced sediment.

37. Line 328 - a shift towards long-term drying – Started much earlier ~10 Ma in southern China region. See Clift et al. (2014), Clift (2025)

We included these references in the manuscript.

38. Line 330 - reflecting enhanced sediment export – You don't know that. Sedimentation rate at a single site can be controlled by a wide range of processes.

We edited "reflecting" to "corresponding to". This clarifies that the increase in MAR is related to the increase in sedimentation rate (as supported by sediment provenance and erosion proxies), but does not suggest that MAR itself is a proxy of sedimentation rate.

39. Line 335 - does not appear to track long-term the monsoon drying – Why do you say that?

Monsoon drying occurs near ~8–5 Ma in Figure 8 of Clift et al. (2014), and ~12–9 Ma in Figure 2 of Clift (2025). At ODP Site 1148, Hm/Gt decreases rather abruptly between ~5.4–4.6 Ma, but remains otherwise stable, therefore, we attribute this decrease to a rapid influx of Taiwan-sourced sediment which reaches steady-state rather than to the influence of drying.

40. Line 337 - as it emerged from the Pacific Ocean - Taiwan (or the collision orogeny) has been progressively migrating to the SE. Any increase is because the collision is approaching the drill site.

Please refer to comment #9.

41. Line 344 - tracks the orogenic evolution of Taiwan at both ODP sites - How does it do that? You just argued that Site 1146 is mostly derived from the Pearl River

We have corrected this typo to only include ODP Site 1148.

42. Line 355 - fine-grained material enriched TOC - fine-grained material enriched in TOC?

We have corrected this typo as the reviewer suggested.

43. Line 369 - reflect increased Eurasian clastic influx under conditions of long-term sea-level fall- How do you know this isn't related to stronger erosion driven by a stronger EASM?

While Clift et al. (2014) and Clift (2025) show the onset of variable monsoon at ~5 Ma and ~9 Ma, respectively, we do not see the corresponding increases in fluctuations recorded in the proxy records at either ODP Sites 1146 or 1148 until after ~3 Ma.

44. Figure 6 - Text is too small to read

We have increased the size of Figure 6.

45. Line 382 - The onset of orogenesis in Taiwan at ~5.5 Ma - The onset of deposition by sediment eroded from the Taiwan Mountains at ~5.5 Ma

We edited the wording to "the onset of Taiwan's emergence at ~5.4 Ma.

46. Line 384 indicates significant export of terrestrial sediment - No. As noted about MAR at a single site could be controlled by many processes, such as currents in the SCS.

[Figure]

We have changed "indicates" to "reflects".

47. Line 387 enhanced by intensified sediment export - Or better preserved by rapid burial

We have included rapid burial as the reviewer suggested.

48. Line 405 - confined to proximal coastal environments - Or oxidized before deposition which seems unlikely compared to the Pearl River

We agree that oxidation before deposition seems unlikely, especially given such high sedimentation rates that organic carbon should be buried (and preserved) rapidly.

49. Line 417 rapidly declining offshore due to swift uptake - And dilution?

Carbon accumulation rates suggest that organic carbon is not diluted by high sediment influx.

50. Line 424 - makes the northern SCS a depocenter for organic carbon burial - Not too much based on this regional synthesis

Clift, P. D., T. N. Jonell, Y. Du, and T. Bornholdt (2024), The impact of Himalayan-Tibetan erosion on silicate weathering and organic carbon burial, Chem. Geol., 656, 122106, doi:10.1016/j.chemgeo.2024.122106.

The northern SCS is indeed a smaller carbon depocenter compared to systems dominated by large deltas, but locally, it plays a significant role in organic carbon accumulation.

51. Line 430 - nutrient inputs from Taiwan and the Yangtze River, and presumably the rivers of Southeast China as well

We edited the wording to indicate that the inputs are *dominantly* from Taiwan and the Yangtze River.

52. Line 445 - Cholan Fm - Don't abbreviate Fm

Although the abbreviation 'fm' is commonly used in geology, we will use the full term 'formation' due to the audience of *Climate of the Past*.

53. Line 450 - The enhanced in export of coarser-grained sediment - The enhanced export of coarser-grained sediment

We have corrected this typo.

54. Line 453 - would in turn increase sediment supply to the South China Sea - 453 would in turn have increased sediment supply to the South China Sea - I'm not sure I understand the logic. Also, you were abbreviating South China Sea earlier in the paper.

We have restructured this section to better communicate our rationale.

55. Line 459 - coupled with intensified monsoon – are they really intensified? I thought it was mostly during the interglacial periods when the monsoon is strong.

Several studies (referenced in the manuscript, lines 448–449) suggest that the EASM intensified near 3.5 Ma, though consensus is lacking. For this reason, we have revised the section to emphasize the combined influence of sea-level fall and precipitation on sediment export, without asserting that the EASM intensified at this time (please see comment #54).

56. Line 467- South China Sea (SCS) – you've already defined this abbreviation

We purposely redefine abbreviations in section 6 to consider readers who may choose to read the conclusions first.

57. Line 480 the East Asian Summer Monsoon - you were abbreviating that before

[Figure]

Please see comment #56.

58. Line 488- rather than to local tectonics - do you mean in Taiwan or are you talking about near the drill sites in the South China Sea?

We clarified this to be Taiwan's orogenesis.

[Figure]

Reviewer #2 (Prof. Shannon Dulin)

The manuscript offers a comparison between Neogene strata sampled from ODP drillsites in the SCS and outcrops of the Taiwan Western Foreland Basin. The data focus on TOC, isotopic C signatures, and mineralogy to determine climatic trends and provenance. This study suggests that increased marine productivity is recorded in ODP sites that show slight increases in TOC and C isotopic values. These values are correlated with increased typhoon activity that mobilized more sediment from Taiwan post-emergence. This is an interesting conclusion that increased weathering caused by enhanced monsoonal circulation and typhoon activity is recorded in the ODP site 1148. The three sites (2 ODP and the combined TWFB outcrops) show distinct (though nuanced and in some cases very slight) differences in the climatic indicators, providing a good comparison for the author's hypothesis. Some of the trends seem slight, and may benefit from a more rigorous statistical analysis, although I think this is fairly compelling presentation as-is. It is on its way to being a good comprehensive study.

**General comments:**

60. The manuscript would benefit greatly from pictures of the lithologies recorded in the numerous outcrop samples (N=1000+), as well as a correlation between the ODP cores and the TWFB locations.

We have added a photo of the outcrop section of the Chinshui Shale at Tachia River.

We did not include a figure correlating the geochemical records between the ODP sites and the TWFB locations due to the large number of datasets. Rather, we showed the chronostratigraphy of the TWFB on Figures 3 and 4 so readers can see the relationship between different stages of Taiwan's orogenesis on these sediment records.

61. The magnetostratigraphy would benefit from rock magnetic analysis to confirm the presence/mineralogy of the ChRM carriers. The NRMs are very high for shales, presumably due to the Fe-rich arc sediments, with unblocking temps indicating greigite/pyrrhotite. A cross-section showing the magnetostratigraphic correlations would strengthen this portion of the study significantly. Some MAD angles are relatively high (>10, samples (e.g. 27B.1), were these excluded from the analysis?

We did not perform rock magnetic analysis for the presence and mineralogy of ChRM carriers because the primary purpose for rock magnetism in this study is as a tool for determining stratal ages and assessing the extent of erosion. Instead, we referred to Horng and Huh (2011) for insight into the provenance of different magnetic minerals exported to the study area. We did not directly show the magnetostratigraphic correlations, but rather, we showed the chronostratigraphic log of the TWFB on both Figures 3, and 4, and the data from the ODP sites are plot against this to show their age-equivalence (please see comment #60). Despite the name, parts of the Chinshui Shale had high sand content, resulting in some samples yielding unreliable paleomagnetism results. The magnetic polarity reversals were determined based on measurements from robust samples, while results from samples with higher potential for error were flagged. Additionally, there is already an existing astronomically tuned age model for the Chinshui Shale, therefore, the magnetobiostratigraphic model was used only to confirm and support the existing age model.

62. Is ODP site 1148 on the continental rise or on the distal shelf?

ODP Site 1148 is located on the continental rise.

**Specific Comments:**

63. Figure 2: Should the Chinshui shale box be grey? What is the purpose of the red box…just highlighting the Kueichulin Fm because of the majority of samples there?

We have edited the colour of the Chinshui Shale to grey and adjusted the red box to highlight the studied formations.

[Figure]

64. Lines ~169-171; Lines189-193: not methods, more discussion/results

We believe these sections are relevant to the methodology because lines 169–171 justify the use of magnetic susceptibility as a sediment provenance indicator in our study, and lines 189–193 justify the use of Hm/Gt as both monsoon and provenance proxy.

65. Age Models: The orthogonal projection diagrams in the supplemental material should show magnetization steps labeled directly on them.

Figure S1: We have included labels for the magnetizations steps as Prof. Dulin suggested.

66. Line 215: in-phase

We have corrected this typo.

67. Line ~243-258: Many of these errors in mag sus overlap…are the trends identified here statistically significant?

While there are some overlaps in the standard deviation in our data, the trends are visually consistent and persistent across numerous measurements, suggesting that the observed patterns reflect genuine changes in the magnetic susceptibility trends rather than random variability.

68. Line 291: no comma after strait; funneling (sp)

We have corrected these typos.

69. Figure 5 is very good and found it extremely useful throughout the manuscript. It may benefit from bathymetry data…the depth difference between sites 1146 and 1148 is significant but does not appear so on first glance of the figure.

Figure 5: We have included the bathymetric map as Prof. Dulin suggested.

70. Line 305: consider adding ~5.4 Ma after emergence.

We have added ~5.4 Ma as Prof. Dulin suggested.

71. Line 308: no hyphen in tropical-cyclone.

We hyphenated tropical-cyclone as it is a compound adjective describing precipitation.

72. Line 330: add "of" between erosion and the

We have added "of".

73. Line 335: remove "the" after long-term

We have removed "the".

74. Line 336: depleted? Or just naturally absent?

Hematite is depleted in Taiwan-sourced sediment (Horng and Huh, 2011).

75. Line 343: add "the" before Taiwan Warm Current

We have added "the".

76. Line 356: add "in" before TOC

We have added "in".

77. Figure 6 is very well presented and convincing for your argument of increased productivity seen particularly in OCP site 1148.
78. Lines ~404-407. Well written and convincing arguments here.

[Figure]

**University of Lausanne**
**Institute of Earth Sciences**
Unil-Mouline, Géopolis
Route de la Mouline 11
1015 Lausanne

We thank Prof. Dulin for these comments.

79. Line 466: remove hypen in late-Miocene

We hyphenated late-Miocene as it is a compound adjective describing the sedimentary/geochemical records.

80. Line 492: add matter or carbon (?) after organic

We have added "carbon".

81. Line 497: add s on tectonic

We have corrected this typo.

[Figure]

[Figure]

Figure 2: A) Map of the study area showing the locations of the Late Miocene–Early Pleistocene records from Ocean Drilling Program (ODP) sediment cores (orange circles) in the South China Sea, and outcrop locations from the Taiwan Western Foreland Basin (TWFB, blue stars). The inset map outlined in blue show the locations of the borehole (HYS-1) and outcrop locations (DRK = Da'an River, Kueichulin Formation; TRC = Tachia River, Chinshui Shale; HRC = Houlong River, Cholan Formation) of the TWFB strata used in this study. Modern-day circulation in the SCS is shown in arrows: black = alongshore surface current, green = surface- and intermediate-water currents, yellow = deep- and bottom-water current, pink = Kuroshio current, pink (dashed) = Taiwan warm current (modified from Hu et al. (2010); Liu et al. (2010); Liu et al. (2016); Yin et al. (2023)). B) Chronostratigraphy of the TWFB is modified after Chen (2016), Hsieh et al. (2023), and Teng et al. (1991). The red box highlights the targeted study section. Yellow denotes sandstone-dominated strata, and grey indicates mudstone-dominated strata. C) Outcrop photo of the Chinshui Shale at Tachia River (this study).

[Figure]

**University of Lausanne**
**Institute of Earth Sciences**
Unil-Mouline, Géopolis
Route de la Mouline 11
1015 Lausanne

[Figure]

Figure 3: Compilation of total organic carbon (TOC), C/N, δ13Corg, and gamma ray data for the Taiwan Western Foreland Basin (TWFB), including the Kueichulin Formation (Dashtgard et al., 2021; Hsieh et al., 2023b; Hsieh et al., 2023a), the Chinshui Shale (this study and gamma-ray from Vaucher et al. (2023b)), and the Cholan Formation (this study and gamma-ray from Vaucher et al. (2023b)). Sea-level curves are from Haq and Ogg (2024). ">" indicates data that plot outside of the diagram. The solid lines represent curves fitted using locally estimated scatterplot smoothing (LOESS). TOC, C/N, and δ13Corg trends reflect organic carbon sources, and show that marine organic matter content is high in the Kuantaoshan Sandstone, Shihliufen Shale, and Chinshui Shale, contrasting with increased terrestrial input in the Yutengping Sandstone and Cholan Formation. Gamma-ray data indicate lithological variability, and correlate with sea-level changes.

[Figure]

**University of Lausanne**
**Institute of Earth Sciences**
Unil-Mouline, Géopolis
Route de la Mouline 11
1015 Lausanne

[Figure]

Figure 4: Compilation of sediment core data from ODP Sites 1146 and 1148 in the northern South China Sea, including mass accumulation rate (MAR; Wan et al., 2010a; Wang et al., 2000a), TOC and δ13Corg (this study), mass-specific magnetic susceptibility (χ; Wang et al., 2005a; Wang et al., 2000a), hematite/goethite (Hm/Gt; Wang et al., 2000b; Clift, 2006), gamma ray (Wang et al., 2000b, a), and Ti/Ca (Wan et al., 2010a; Hoang et al., 2010). Sea-level curves are from Haq and Ogg (2024). ">" indicates data that plot outside of the diagram. The solid lines represent curves fitted using locally estimated scatterplot smoothing (LOESS). The figure illustrates the contrasting sedimentary and geochemical responses between the two ODP sites, driven by tectonic uplift, climate variability, and changes in ocean circulation.

[Figure]

[Figure]

Figure 5: Summary of different controls on sediment and carbon accumulation over time in the Taiwan Western Foreland Basin (blue star) and the ODP sites (orange circles) in the northern South China Sea. The size of the arrows indicates relative proportions of sediment flux, and brown indicates accumulation of terrestrial organic carbon, while yellow indicates marine organic carbon. The abbreviations MT = Manilla Trench, RT = Ryukyu Trench, and PT = proto-Taiwan. These differences in organic carbon source (i.e., terrestrial vs. marine) and carbon accumulation highlight the spatial heterogeneity in sedimentary and geochemical records within the northern South China Sea, shaped by the interplay of tectonic and climatic processes. Bathymetric map from Gebco Compilation Group, 2025).

[Figure]

Figure 6: Cross-plot of δ13Corg and TOC measured from ODP Sites 1146 and 1148. Values are grouped according to major tectonic and climate changes: 1) pre-emergence of Taiwan and pre-Northern Hemisphere Glaciation, 2) post-emergence of Taiwan and pre-Northern Hemisphere Glaciation, and 3) post-emergence of Taiwan and post-Northern Hemisphere Glaciation. Note the distinct trends before and after Taiwan's emergence and Northern Hemisphere Glaciation. Site 1146 reflects Eurasian sediment input with marine organic matter dominance, while Site 1148 highlights Taiwan's influence, with enhanced marine productivity linked to nutrient export.

[Figure]

**Sample: 1B.1**

PCA  dec 1.70 / inc 51.63
PCA  MAD1 36.50 / MAD3 4.44
(0.62 0.02 0.78)t

| | temp. | dec. | inc. | int. | m.s. |
|---|---|---|---|---|---|
| * | 25 | 1.6 | 52.9 | 1.43e−02 | 2.8e01 |
| * | 120 | 357.0 | 50.6 | 1.21e−02 | 2.8e01 |
| * | 160 | 4.2 | 54.3 | 6.27e−03 | 2.9e01 |
| * | 200 | 2.4 | 47.6 | 5.54e−03 | 2.9e01 |
| * | 240 | 0.8 | 50.3 | 5.15e−03 | 2.8e01 |
| * | 280 | 11.5 | 44.7 | 4.87e−03 | 2.6e01 |
| * | 320 | 20.5 | 50.6 | 3.58e−03 | 2.7e01 |
| * | 360 | 42.0 | 64.3 | 2.37e−03 | 2.7e01 |
| | 400 | 211.8 | −25.9 | 8.31e−04 | 2.8e01 |
| | 440 | 144.7 | −11.5 | 1.33e−03 | 3.3e01 |
| | 480 | 226.7 | 33.3 | 1.24e−03 | 3.4e01 |

[Figure]

**Figure S1** **A)** Thermal demagnetization steps showing intensity of remanent magnetization (■), and magnetic susceptibility (□), **B)** stereographic (■ = positive inclination values, □ = negative inclination values), and **C)** orthogonal plots for samples analysed for paleomagnetism. Temp. = temperature step, dec. = declination, inc. = inclination, int. = intensity of remanent magnetization, and and m.s. = magnetic susceptibility. Stable magnetic remanences (*, data points highlighted in red) were used for principal component analysis, and the resulting characteristic remanent components (ChRM declination and inclination) toward the origin point are shown with blue lines. Anomalously high remanent magnetization intensities were removed from the plots (-) for clarity.

**Sample: 2A.1**

[Figure]

PCA dec 12.82 / inc 44.57
PCA MAD1 19.88 / MAD3 3.44
(0.69 0.16 0.70)t

| | temp. | dec. | inc. | int. | m.s. |
|---|---|---|---|---|---|
| * | 25 | 16.0 | 44.0 | 1.61e−02 | 3.3e01 |
| * | 120 | 10.5 | 45.9 | 1.16e−02 | 3.2e01 |
| * | 160 | 9.7 | 40.8 | 5.42e−03 | 3.2e01 |
| * | 200 | 9.4 | 45.4 | 4.79e−03 | 3.2e01 |
| * | 240 | 356.6 | 44.9 | 4.80e−03 | 3.2e01 |
| * | 280 | 13.3 | 44.1 | 4.13e−03 | 3.0e01 |
| | 320 | 58.1 | 66.2 | 3.41e−03 | 3.0e01 |
| | 360 | 69.1 | 51.1 | 2.02e−03 | 3.0e01 |
| | 400 | 56.3 | 19.9 | 3.84e−03 | 3.7e01 |
| | 440 | 29.2 | 41.3 | 4.18e−03 | 5.1e01 |
| | 480 | 18.9 | 42.6 | 2.72e−03 | 6.0e01 |

[Figure]

**Sample: 3A.1**

□ vertical
■ horizontal
Units: A/m ×10$^{-3}$

PCA  dec 14.40 / inc 51.92
PCA  MAD1 28.19 / MAD3 7.65
(0.60 0.15 0.79)t

| | temp. | dec. | inc. | int. | m.s. |
|---|---|---|---|---|---|
| * | 25 | 12.3 | 56.9 | 1.55e−02 | 3.0e01 |
| * | 120 | 11.9 | 50.5 | 1.10e−02 | 3.0e01 |
| * | 160 | 27.6 | 44.0 | 6.21e−03 | 3.0e01 |
| * | 200 | 17.3 | 39.8 | 4.50e−03 | 3.0e01 |
| * | 240 | 27.1 | 37.6 | 3.78e−03 | 2.9e01 |
| * | 280 | 20.7 | 36.4 | 3.83e−03 | 2.8e01 |
| * | 320 | 356.4 | 37.6 | 3.33e−03 | 2.7e01 |
| | 360 | 333.2 | 26.4 | 2.20e−03 | 2.8e01 |
| | 400 | 293.0 | 25.1 | 1.66e−03 | 3.8e01 |
| | 440 | 346.4 | 11.8 | 1.82e−03 | 5.8e01 |
| | 480 | 54.2 | −69.1 | 1.35e−03 | 7.0e01 |

[Figure]

**Sample: 4A.1**

□ vertical
■ horizontal
Units: A/m ×10$^{-3}$

PCA  dec 3.92 / inc 0.17
PCA  MAD1 27.82 / MAD3 7.25
(1.00 0.07 0.00)t

| | temp. | dec. | inc. | int. | m.s. |
|---|---|---|---|---|---|
| * | 25 | 359.6 | 3.5 | 1.62e−02 | 3.5e01 |
| * | 120 | 3.0 | −1.3 | 1.21e−02 | 3.4e01 |
| * | 160 | 12.9 | −1.1 | 8.93e−03 | 3.4e01 |
| * | 200 | 16.9 | −1.9 | 7.84e−03 | 3.4e01 |
| * | 240 | 357.8 | −7.2 | 5.44e−03 | 3.3e01 |
| * | 280 | 1.4 | −12.0 | 4.17e−03 | 3.3e01 |
| * | 320 | 358.4 | 4.9 | 2.45e−03 | 3.1e01 |
| * | 360 | 14.7 | −9.6 | 2.14e−03 | 3.2e01 |
| | 400 | 61.3 | −50.2 | 1.62e−03 | 4.7e01 |
| | 440 | 93.5 | −45.5 | 1.75e−03 | 7.6e01 |
| | 480 | 102.2 | −33.2 | 5.39e−03 | 1.0e02 |

[Figure]

**Sample: 5A.1**

□ vertical
■ horizontal
Units: A/m ×10$^{-3}$

PCA  dec 22.84 / inc 40.88
PCA  MAD1 26.22 / MAD3 8.13
(0.70 0.29 0.65)t

| | temp. | dec. | inc. | int. | m.s. |
|---|---|---|---|---|---|
| * | 25 | 22.9 | 45.0 | 1.38e−02 | 3.2e01 |
| * | 120 | 19.0 | 41.2 | 8.92e−03 | 3.0e01 |
| * | 160 | 25.7 | 39.4 | 4.96e−03 | 3.1e01 |
| * | 200 | 27.1 | 33.6 | 4.86e−03 | 3.7e01 |
| * | 240 | 14.5 | 26.2 | 4.18e−03 | 3.0e01 |
| * | 280 | 34.2 | 14.2 | 3.41e−03 | 2.9e01 |
| * | 320 | 47.2 | 44.4 | 2.17e−03 | 2.6e01 |
| * | 360 | 27.4 | 20.0 | 2.68e−03 | 2.9e01 |
| * | 400 | 30.4 | −8.7 | 4.21e−04 | 4.3e01 |
| | 440 | 263.2 | 57.0 | 1.50e−03 | 6.9e01 |
| | 480 | 290.1 | 35.1 | 3.82e−03 | 7.9e01 |

[Figure]

**Sample: 6B.1**

□ vertical
■ horizontal
Units: A/m ×10$^{-3}$

PCA  dec 23.67 / inc 68.46
PCA  MAD1 15.10 / MAD3 8.41
(0.34 0.15 0.93)t

| | temp. | dec. | inc. | int. | m.s. |
|---|---|---|---|---|---|
| * | 25 | 8.1 | 68.3 | 1.49e−02 | 3.0e01 |
| * | 120 | 20.9 | 70.8 | 8.95e−03 | 2.9e01 |
| * | 160 | 44.9 | 66.2 | 6.08e−03 | 2.9e01 |
| * | 200 | 39.2 | 65.0 | 5.93e−03 | 3.0e01 |
| * | 240 | 60.1 | 65.8 | 5.14e−03 | 2.9e01 |
| * | 280 | 58.2 | 51.7 | 4.91e−03 | 2.8e01 |
| | 320 | 108.5 | 74.2 | 4.40e−03 | 2.7e01 |
| | 360 | 140.0 | 28.2 | 2.68e−03 | 2.6e01 |
| | 400 | 114.0 | 1.4 | 2.41e−03 | 4.3e01 |
| | 440 | 131.5 | 5.6 | 2.38e−03 | 7.3e01 |
| | 480 | 218.3 | −16.2 | 5.37e−03 | 7.6e01 |

[Figure]

**Sample: 7B.1**

□ vertical
■ horizontal
Units: A/m ×10$^{-3}$

PCA  dec 331.21 / inc 53.38
PCA  MAD1 19.86 / MAD3 5.88
(0.52 −0.29 0.80)t

| | temp. | dec. | inc. | int. | m.s. |
|---|---|---|---|---|---|
| * | 25 | 327.3 | 52.5 | 1.35e−02 | 2.7e01 |
| * | 120 | 328.6 | 55.5 | 7.48e−03 | 2.6e01 |
| * | 160 | 346.6 | 49.8 | 4.83e−03 | 2.6e01 |
| * | 200 | 344.6 | 55.5 | 3.67e−03 | 2.6e01 |
| * | 240 | 355.0 | 62.1 | 2.15e−03 | 2.6e01 |
| * | 280 | 10.5 | 46.6 | 2.54e−03 | 2.5e01 |
| | 320 | 325.5 | 30.6 | 2.66e−03 | 2.4e01 |
| | 360 | 330.9 | 1.4 | 3.30e−03 | 2.5e01 |
| | 400 | 0.5 | −19.7 | 3.40e−03 | 4.2e01 |
| | 440 | 7.5 | 34.2 | 2.04e−03 | 7.8e01 |
| | 480 | 96.7 | 32.5 | 7.56e−03 | 8.8e01 |

[Figure]

**Sample: 8A.1**

□ vertical
■ horizontal
Units: A/m ×10$^{-3}$

PCA  dec 359.71 / inc 36.45
PCA  MAD1 24.71 / MAD3 10.88
(0.80 −0.00 0.59)t

| | temp. | dec. | inc. | int. | m.s. |
|---|---|---|---|---|---|
| * | 25 | 350.8 | 42.3 | 1.21e−02 | 2.1e01 |
| * | 120 | 6.2 | 40.5 | 7.19e−03 | 2.1e01 |
| * | 160 | 9.8 | 35.1 | 5.79e−03 | 2.0e01 |
| * | 200 | 2.4 | 28.4 | 5.10e−03 | 2.0e01 |
| * | 240 | 5.6 | 29.9 | 4.82e−03 | 2.0e01 |
| * | 280 | 8.6 | 21.4 | 4.46e−03 | 1.9e01 |
| * | 320 | 357.9 | 21.0 | 3.97e−03 | 1.9e01 |
| * | 360 | 23.7 | 13.0 | 1.85e−03 | 2.0e01 |
| * | 400 | 9.5 | 5.8 | 2.57e−03 | 2.8e01 |
| | 440 | 43.1 | 51.8 | 2.54e−03 | 4.6e01 |
| | 480 | 141.0 | 4.0 | 7.99e−03 | 4.5e01 |

[Figure]

**Sample: 9A.1**

□ vertical
■ horizontal
Units: A/m ×10$^{-2}$

PCA  dec 357.82 / inc 39.34
PCA  MAD1 13.64 / MAD3 10.91
(0.77 −0.03 0.63)t

| | temp. | dec. | inc. | int. | m.s. |
|---|---|---|---|---|---|
| * | 25 | 359.7 | 44.1 | 1.23e−02 | 3.1e01 |
| * | 120 | 352.9 | 33.6 | 6.13e−03 | 3.0e01 |
| * | 160 | 355.5 | 29.1 | 3.64e−03 | 3.0e01 |
| * | 200 | 10.0 | 26.8 | 2.34e−03 | 3.0e01 |
| * | 240 | 346.1 | 3.1 | 2.54e−03 | 2.9e01 |
| * | 280 | 351.5 | −9.8 | 2.24e−03 | 2.8e01 |
| | 320 | 324.3 | 32.6 | 1.04e−03 | 2.6e01 |
| | 360 | 12.1 | 76.1 | 1.91e−03 | 2.8e01 |
| | 400 | 330.1 | 14.5 | 2.01e−03 | 4.4e01 |
| | 440 | 272.6 | −6.6 | 1.03e−02 | 8.6e01 |
| | 480 | 290.3 | 0.6 | 5.11e−02 | 1.0e02 |

[Figure]

**Sample: 10B.1**

□ vertical
■ horizontal
Units: A/m ×10$^{-3}$

PCA  dec 188.31 / inc 1.95
PCA  MAD1 30.33 / MAD3 7.42
(−0.99 −0.14 0.03)t

| | temp. | dec. | inc. | int. | m.s. |
|---|---|---|---|---|---|
| * | 25 | 187.0 | 1.9 | 1.16e−02 | 2.9e01 |
| * | 120 | 196.4 | 2.3 | 5.98e−03 | 2.8e01 |
| * | 160 | 189.7 | −2.0 | 5.08e−03 | 2.8e01 |
| * | 200 | 187.6 | −0.7 | 4.36e−03 | 2.8e01 |
| * | 240 | 182.4 | 31.6 | 2.78e−03 | 2.7e01 |
| * | 280 | 180.9 | 10.9 | 2.81e−03 | 2.6e01 |
| * | 320 | 182.6 | −16.1 | 2.66e−03 | 2.8e01 |
| | 360 | 157.3 | −29.0 | 1.64e−03 | 2.7e01 |
| | 400 | 161.8 | −64.8 | 3.16e−03 | 4.8e01 |
| - | 440 | 239.9 | −29.1 | 3.07e−02 | 1.3e02 |
| - | 480 | 309.5 | 1.0 | 2.24e−01 | 1.7e02 |

[Figure]

**Sample: 11A.1**

□ vertical
■ horizontal
Units: A/m ×10$^{-3}$

PCA  dec 222.58 / inc −11.87
PCA  MAD1 12.56 / MAD3 6.49
(−0.72 −0.66 −0.21)t

| | temp. | dec. | inc. | int. | m.s. |
|---|---|---|---|---|---|
| | 25 | 227.0 | 13.5 | 1.12e−02 | 2.4e01 |
| | 120 | 228.3 | 4.9 | 5.71e−03 | 2.4e01 |
| | 160 | 232.9 | 3.2 | 3.90e−03 | 2.3e01 |
| | 200 | 231.9 | 1.1 | 3.49e−03 | 2.4e01 |
| * | 240 | 221.0 | −11.6 | 5.00e−03 | 2.2e01 |
| * | 280 | 233.6 | −12.3 | 3.48e−03 | 2.2e01 |
| * | 320 | 216.1 | −10.9 | 3.47e−03 | 2.3e01 |
| * | 360 | 213.6 | −19.5 | 1.18e−03 | 2.2e01 |
| | 400 | 137.2 | −33.0 | 3.91e−03 | 4.0e01 |
| - | 440 | 23.8 | 19.4 | 5.57e−02 | 6.9e01 |
| - | 480 | 92.4 | 22.2 | 1.23e−01 | 6.5e01 |

[Figure]

**Sample: 12B.1**

□ vertical
■ horizontal
Units: A/m ×10$^{-3}$

PCA dec 14.17 / inc 10.28
PCA MAD1 29.45 / MAD3 23.70
(0.95 0.24 0.18)t

| | temp. | dec. | inc. | int. | m.s. |
|---|---|---|---|---|---|
| | 25 | 15.4 | 69.5 | 1.20e−02 | 2.8e01 |
| | 120 | 35.0 | 62.7 | 6.73e−03 | 2.7e01 |
| | 160 | 39.0 | 56.6 | 4.38e−03 | 2.7e01 |
| | 200 | 80.0 | 56.6 | 2.18e−03 | 2.8e01 |
| * | 240 | 5.8 | 22.4 | 2.54e−03 | 2.6e01 |
| * | 280 | 24.6 | 20.7 | 3.43e−03 | 2.5e01 |
| * | 320 | 36.0 | −9.4 | 1.67e−03 | 2.5e01 |
| * | 360 | 6.9 | 25.3 | 1.56e−03 | 2.6e01 |
| * | 400 | 355.3 | −24.4 | 2.73e−03 | 4.2e01 |
| - | 440 | 42.4 | 8.6 | 3.83e−02 | 7.7e01 |
| - | 480 | 31.3 | 7.4 | 6.75e−02 | 7.6e01 |

[Figure]

**Sample: 13A.1**

□ vertical
■ horizontal
Units: A/m ×10$^{-3}$

PCA  dec 179.96 / inc 50.03
PCA  MAD1 32.86 / MAD3 9.43
(−0.64 0.00 0.77)t

| | temp. | dec. | inc. | int. | m.s. |
|---|---|---|---|---|---|
| * | 25 | 173.0 | 52.2 | 1.32e−02 | 3.3e01 |
| * | 120 | 180.6 | 43.7 | 7.73e−03 | 3.2e01 |
| * | 160 | 196.4 | 46.5 | 5.57e−03 | 3.2e01 |
| * | 200 | 204.4 | 44.2 | 4.31e−03 | 3.2e01 |
| * | 240 | 174.9 | 75.6 | 3.41e−03 | 3.0e01 |
| * | 280 | 188.0 | 35.6 | 3.03e−03 | 2.9e01 |
| * | 320 | 194.9 | 29.1 | 1.79e−03 | 3.0e01 |
| | 360 | 254.8 | −23.2 | 1.54e−03 | 3.1e01 |
| | 400 | 280.8 | −66.3 | 1.84e−03 | 4.4e01 |
| - | 440 | 150.0 | −21.4 | 1.96e−02 | 8.9e01 |
| - | 480 | 102.6 | −9.7 | 2.68e−02 | 8.5e01 |

[Figure]

**Sample: 14A.1**

□ vertical
■ horizontal
Units: A/m ×10$^{-3}$

PCA  dec 178.53 / inc 9.28
PCA  MAD1 31.78 / MAD3 6.39
(−0.99 0.03 0.16)t

| | temp. | dec. | inc. | int. | m.s. |
|---|---|---|---|---|---|
| * | 25 | 177.7 | 7.7 | 1.02e−02 | 3.3e01 |
| * | 120 | 180.4 | 7.1 | 5.24e−03 | 3.3e01 |
| * | 160 | 171.5 | 19.9 | 4.09e−03 | 3.2e01 |
| * | 200 | 179.1 | 13.6 | 3.74e−03 | 3.3e01 |
| * | 240 | 177.9 | 11.0 | 2.11e−03 | 3.2e01 |
| * | 280 | 198.7 | 1.5 | 2.48e−03 | 3.2e01 |
| * | 320 | 198.8 | 22.1 | 1.34e−03 | 2.9e01 |
| | 360 | 272.3 | −26.5 | 1.15e−03 | 3.6e01 |
| | 400 | 195.5 | −80.3 | 1.93e−03 | 5.0e01 |
| - | 440 | 207.7 | −37.3 | 1.36e−02 | 1.3e02 |
| - | 480 | 339.2 | −27.9 | 2.37e−02 | 1.6e02 |

[Figure]

**Sample: 15A.1**

□ vertical
■ horizontal
Units: A/m ×10$^{-3}$

PCA  dec 1.29 / inc 43.70
PCA  MAD1 13.87 / MAD3 7.10
(0.72 0.02 0.69)t

| | temp. | dec. | inc. | int. | m.s. |
|---|---|---|---|---|---|
| * | 25 | 6.1 | 50.3 | 2.65e−02 | 4.2e01 |
| * | 120 | 1.3 | 43.9 | 2.01e−02 | 4.1e01 |
| * | 160 | 357.4 | 41.9 | 1.81e−02 | 4.0e01 |
| * | 200 | 4.2 | 43.2 | 1.54e−02 | 4.1e01 |
| * | 240 | 358.7 | 37.8 | 1.37e−02 | 4.0e01 |
| * | 280 | 351.7 | 33.4 | 1.13e−02 | 4.0e01 |
| * | 320 | 359.4 | 41.0 | 8.05e−03 | 3.9e01 |
| * | 360 | 1.4 | 32.5 | 6.53e−03 | 4.5e01 |
| * | 400 | 356.2 | 18.7 | 7.24e−03 | 6.4e01 |
| | 440 | 333.8 | −2.1 | 1.34e−02 | 2.7e02 |
| | 480 | 328.4 | −14.0 | 2.27e−02 | 3.0e02 |

[Figure]

**Sample: 16A.2**

□ vertical
■ horizontal
Units: A/m ×10$^{-3}$

PCA  dec 117.50 / inc −0.47
PCA  MAD1 29.29 / MAD3 9.77
(−0.46 0.89 −0.01)t

| | temp. | dec. | inc. | int. | m.s. |
|---|---|---|---|---|---|
| | 25 | 83.7 | 17.9 | 1.23e−02 | 2.0e01 |
| | 120 | 90.2 | 19.6 | 7.33e−03 | 2.0e01 |
| * | 160 | 108.2 | 3.8 | 2.99e−03 | 2.0e01 |
| * | 200 | 119.8 | 7.6 | 2.60e−03 | 1.9e01 |
| * | 240 | 116.5 | 7.3 | 2.18e−03 | 1.9e01 |
| * | 280 | 120.9 | 2.8 | 2.32e−03 | 1.8e01 |
| * | 320 | 126.3 | −12.9 | 2.76e−03 | 1.8e01 |
| * | 360 | 117.5 | −3.2 | 2.10e−03 | 1.8e01 |
| * | 400 | 115.0 | −10.2 | 2.11e−03 | 1.8e01 |
| | 440 | 189.7 | 11.7 | 1.43e−03 | 2.9e01 |
| | 480 | 168.6 | 12.4 | 1.77e−03 | 3.2e01 |

[Figure]

[Figure]

**Sample: 17B.1**

□ vertical
■ horizontal
Units: A/m ×10$^{-3}$

PCA  dec 357.82 / inc 55.27
PCA  MAD1 35.72 / MAD3 7.27
(0.57 −0.02 0.82)t

| | temp. | dec. | inc. | int. | m.s. |
|---|---|---|---|---|---|
| * | 25 | 351.8 | 56.1 | 1.71e−02 | 2.9e01 |
| * | 120 | 348.7 | 52.1 | 1.19e−02 | 2.9e01 |
| * | 160 | 7.6 | 54.9 | 8.14e−03 | 2.9e01 |
| * | 200 | 8.2 | 55.3 | 7.06e−03 | 2.9e01 |
| * | 240 | 15.0 | 56.3 | 5.77e−03 | 2.8e01 |
| * | 280 | 16.5 | 52.9 | 5.06e−03 | 2.8e01 |
| * | 320 | 3.9 | 57.4 | 4.06e−03 | 2.8e01 |
| * | 360 | 10.0 | 55.6 | 4.98e−03 | 2.7e01 |
| * | 400 | 13.7 | 39.4 | 4.98e−03 | 2.7e01 |
| * | 440 | 345.4 | 53.8 | 4.31e−03 | 5.0e01 |
| * | 480 | 24.3 | 71.5 | 4.39e−03 | 6.4e01 |

[Figure]

**Sample: 18B**

[Figure]

PCA  dec 356.38 / inc 35.86
PCA  MAD1 5.53 / MAD3 12.68
(0.81 −0.05 0.59)t

| | temp. | dec. | inc. | int. | m.s. |
|---|---|---|---|---|---|
| * | 25 | 9.2 | 45.8 | 5.64e−02 | 1.3e02 |
| * | 120 | 349.0 | 29.2 | 4.33e−02 | 1.3e02 |
| * | 160 | 349.3 | 27.5 | 3.14e−02 | 1.3e02 |
| * | 200 | 348.3 | 26.3 | 2.31e−02 | 1.3e02 |
| * | 240 | 344.0 | 18.8 | 1.71e−02 | 1.3e02 |
| * | 280 | 347.5 | 18.2 | 1.46e−02 | 1.3e02 |
| * | 320 | 344.0 | 35.1 | 8.08e−03 | 1.3e02 |
| | 360 | 23.5 | 28.0 | 5.24e−03 | 1.3e02 |
| | 400 | 357.6 | −7.2 | 4.87e−03 | 1.3e02 |
| | 440 | 317.5 | 20.2 | 5.32e−03 | 1.5e02 |
| | 480 | 320.1 | −34.9 | 6.23e−03 | 1.8e02 |

vertical
horizontal
Units: A/m ×10$^{-2}$

**Sample: 19A**

□ vertical
■ horizontal
Units: A/m ×10$^{-3}$

PCA  dec 15.17 / inc 46.46
PCA  MAD1 5.26 / MAD3 8.54
(0.66 0.18 0.72)t

| | temp. | dec. | inc. | int. | m.s. |
|---|---|---|---|---|---|
| * | 25 | 8.6 | 48.6 | 2.53e−02 | 6.1e01 |
| * | 120 | 14.0 | 47.3 | 1.60e−02 | 6.1e01 |
| * | 160 | 21.8 | 41.6 | 1.05e−02 | 6.0e01 |
| * | 200 | 32.7 | 37.6 | 7.79e−03 | 6.0e01 |
| * | 240 | 38.3 | 33.4 | 6.51e−03 | 5.8e01 |
| * | 280 | 45.7 | 31.4 | 6.31e−03 | 5.8e01 |
| | 320 | 62.0 | 29.2 | 5.40e−03 | 5.6e01 |
| | 360 | 76.2 | 41.6 | 3.80e−03 | 5.4e01 |
| | 400 | 120.5 | 16.3 | 2.67e−03 | 5.4e01 |
| | 440 | 47.9 | 56.0 | 1.53e−03 | 8.8e01 |
| | 480 | 6.2 | 46.1 | 2.06e−03 | 1.2e02 |

[Figure]

**Sample: 20B.1**

□ vertical
■ horizontal
Units: A/m ×10$^{-3}$

PCA  dec 359.19 / inc 57.01
PCA  MAD1 37.82 / MAD3 4.69
(0.54 −0.01 0.84)t

| | temp. | dec. | inc. | int. | m.s. |
|---|---|---|---|---|---|
| * | 25 | 4.8 | 56.2 | 3.30e−02 | 7.6e01 |
| * | 120 | 352.1 | 54.7 | 2.25e−02 | 7.6e01 |
| * | 160 | 354.5 | 56.3 | 1.58e−02 | 7.5e01 |
| * | 200 | 344.2 | 62.2 | 1.29e−02 | 7.3e01 |
| * | 240 | 4.0 | 62.9 | 9.72e−03 | 7.3e01 |
| * | 280 | 6.6 | 60.8 | 9.14e−03 | 7.2e01 |
| * | 320 | 5.1 | 67.4 | 5.23e−03 | 6.9e01 |
| * | 360 | 352.7 | 47.1 | 4.60e−03 | 6.4e01 |
| | 400 | 3.5 | 14.0 | 2.21e−03 | 6.4e01 |
| | 440 | 158.8 | 41.4 | 5.59e−03 | 1.1e02 |
| | 480 | 187.4 | 31.2 | 6.35e−03 | 1.2e02 |

[Figure]

**Sample: 21A.1**

□ vertical
■ horizontal
Units: A/m ×10$^{-3}$

PCA   dec 0.85 / inc 59.65
PCA   MAD1 22.01 / MAD3 4.20
(0.51 0.01 0.86)t

| | temp. | dec. | inc. | int. | m.s. |
|---|---|---|---|---|---|
| * | 25 | 356.1 | 57.9 | 1.83e−02 | 7.0e01 |
| * | 120 | 6.4 | 60.6 | 1.24e−02 | 6.9e01 |
| * | 160 | 9.5 | 60.5 | 7.77e−03 | 6.9e01 |
| * | 200 | 9.5 | 62.5 | 5.04e−03 | 6.8e01 |
| * | 240 | 0.0 | 67.4 | 3.07e−03 | 6.8e01 |
| * | 280 | 33.9 | 75.7 | 3.16e−03 | 6.6e01 |
| | 320 | 169.3 | 13.5 | 1.42e−03 | 6.3e01 |
| | 360 | 146.1 | 52.1 | 4.00e−03 | 5.7e01 |
| | 400 | 169.8 | 38.2 | 1.39e−03 | 5.8e01 |
| | 440 | 305.0 | −69.2 | 3.33e−03 | 8.7e01 |
| | 480 | 341.3 | −10.2 | 3.88e−03 | 1.2e02 |

[Figure]

**Sample: 22B.1**

□ vertical
■ horizontal
Units: A/m ×10$^{-3}$

PCA  dec 348.47 / inc 55.37
PCA  MAD1 40.12 / MAD3 7.46
(0.56 −0.11 0.82)t

| | temp. | dec. | inc. | int. | m.s. |
|---|---|---|---|---|---|
| * | 25 | 345.4 | 52.7 | 2.50e−02 | 6.3e01 |
| * | 120 | 352.1 | 51.0 | 1.64e−02 | 6.2e01 |
| * | 160 | 329.6 | 61.3 | 1.43e−02 | 6.1e01 |
| * | 200 | 8.2 | 61.4 | 9.96e−03 | 6.1e01 |
| * | 240 | 11.2 | 57.0 | 8.75e−03 | 6.0e01 |
| * | 280 | 4.0 | 62.8 | 7.57e−03 | 5.8e01 |
| * | 320 | 6.3 | 48.3 | 3.18e−03 | 5.5e01 |
| | 360 | 270.3 | 25.4 | 1.14e−03 | 5.2e01 |
| | 400 | 28.2 | −55.1 | 1.80e−03 | 5.3e01 |
| | 440 | 131.8 | −39.4 | 9.42e−03 | 8.7e01 |
| | 480 | 177.8 | −23.9 | 6.02e−03 | 9.6e01 |

[Figure]

**Sample: 23B.1**

□ vertical
■ horizontal
Units: A/m ×10$^{-3}$

PCA  dec 340.19 / inc 39.19
PCA  MAD1 24.42 / MAD3 3.64
(0.73 −0.26 0.63)t

| | temp. | dec. | inc. | int. | m.s. |
|---|---|---|---|---|---|
| * | 25 | 341.9 | 39.8 | 3.09e−02 | 4.8e01 |
| * | 120 | 338.4 | 40.3 | 1.68e−02 | 4.8e01 |
| * | 160 | 335.3 | 31.5 | 1.14e−02 | 4.8e01 |
| * | 200 | 329.6 | 36.0 | 6.73e−03 | 4.4e01 |
| * | 240 | 344.8 | 52.2 | 3.98e−03 | 4.6e01 |
| | 280 | 308.6 | 66.4 | 3.06e−03 | 4.5e01 |
| | 320 | 18.1 | 46.1 | 2.30e−03 | 4.4e01 |
| | 360 | 292.5 | −1.7 | 1.55e−03 | 4.3e01 |
| | 400 | 203.5 | −34.1 | 3.44e−03 | 4.7e01 |
| | 440 | 264.5 | −42.4 | 2.13e−02 | 9.7e01 |
| | 480 | 297.0 | −17.4 | 1.53e−02 | 1.4e02 |

[Figure]

**Sample: 24B.1**

□ vertical
■ horizontal
Units: A/m ×10$^{-3}$

PCA  dec 6.67 / inc 64.06
PCA  MAD1 35.91 / MAD3 5.97
(0.43 0.05 0.90)t

| | temp. | dec. | inc. | int. | m.s. |
|---|---|---|---|---|---|
| * | 25 | 6.5 | 63.8 | 1.89e−02 | 6.1e01 |
| * | 120 | 357.0 | 60.1 | 1.24e−02 | 6.0e01 |
| * | 160 | 13.3 | 70.4 | 8.32e−03 | 5.9e01 |
| * | 200 | 29.5 | 72.8 | 7.38e−03 | 5.8e01 |
| * | 240 | 10.5 | 66.1 | 5.29e−03 | 5.8e01 |
| * | 280 | 355.3 | 61.1 | 6.35e−03 | 5.6e01 |
| * | 320 | 29.5 | 53.4 | 5.85e−03 | 5.6e01 |
| | 360 | 5.5 | 41.4 | 4.66e−03 | 5.2e01 |
| | 400 | 55.5 | 54.6 | 4.65e−03 | 5.8e01 |
| - | 440 | 294.9 | −67.4 | 3.82e−02 | 9.5e01 |
| - | 480 | 321.5 | −11.8 | 3.33e−02 | 1.2e02 |

[Figure]

**Sample: 25B.1**

□ vertical
■ horizontal
Units: A/m ×10$^{-3}$

PCA  dec 329.87 / inc 51.22
PCA  MAD1 11.67 / MAD3 5.94
(0.54 −0.31 0.78)t

|   | temp. | dec. | inc. | int. | m.s. |
|---|-------|------|------|------|------|
| * | 25 | 328.4 | 49.4 | 1.51e−02 | 5.3e01 |
| * | 120 | 337.8 | 59.0 | 7.75e−03 | 5.2e01 |
| * | 160 | 329.7 | 57.4 | 4.20e−03 | 5.1e01 |
| * | 200 | 327.5 | 26.2 | 2.82e−03 | 5.0e01 |
|   | 240 | 28.5 | 84.4 | 1.76e−03 | 5.0e01 |
|   | 280 | 290.7 | 14.1 | 1.66e−03 | 4.7e01 |
|   | 320 | 283.6 | −6.2 | 2.08e−03 | 4.8e01 |
|   | 360 | 248.4 | 38.2 | 2.78e−03 | 4.6e01 |
|   | 400 | 335.4 | 60.5 | 3.02e−03 | 5.2e01 |
| - | 440 | 336.5 | −27.4 | 1.07e−01 | 9.6e01 |
| - | 480 | 353.8 | 12.0 | 7.63e−02 | 1.2e02 |

[Figure]

**Sample: 26A.1**

□ vertical
■ horizontal
Units: A/m ×10$^{-3}$

PCA  dec 2.72 / inc 77.07
PCA  MAD1 2.73 / MAD3 5.22
(0.22 0.01 0.97)t

| | temp. | dec. | inc. | int. | m.s. |
|---|---|---|---|---|---|
| * | 25 | 351.7 | 75.8 | 1.21e−02 | 3.9e01 |
| * | 120 | 20.0 | 78.3 | 6.53e−03 | 3.8e01 |
| * | 160 | 61.4 | 74.0 | 4.34e−03 | 3.8e01 |
| | 200 | 168.5 | 72.1 | 2.99e−03 | 3.7e01 |
| | 240 | 99.4 | 49.1 | 2.19e−03 | 3.6e01 |
| | 280 | 97.3 | 50.7 | 2.00e−03 | 3.4e01 |
| | 320 | 121.4 | 13.4 | 1.87e−03 | 3.5e01 |
| | 360 | 81.7 | −10.3 | 9.63e−04 | 3.6e01 |
| | 400 | 178.5 | −64.9 | 5.87e−03 | 4.5e01 |
| - | 440 | 283.0 | −51.9 | 1.28e−01 | 8.7e01 |
| - | 480 | 277.4 | −56.6 | 9.75e−02 | 1.3e02 |

[Figure]

**Sample: 27B.1**

□ vertical
■ horizontal
Units: A/m ×10$^{-3}$

PCA  dec 202.07 / inc 44.36
PCA  MAD1 31.86 / MAD3 18.66
(−0.66 −0.27 0.70)t

| temp. | dec. | inc. | int. | m.s. |
|---|---|---|---|---|
|   | 25 | 287.5 | 53.8 | 1.53e−02 | 4.6e01 |
|   | 120 | 260.3 | 56.1 | 8.71e−03 | 4.4e01 |
|   | 160 | 250.2 | 52.4 | 5.91e−03 | 4.4e01 |
| * | 200 | 217.5 | 42.5 | 4.73e−03 | 4.4e01 |
| * | 240 | 198.8 | 53.7 | 3.87e−03 | 4.3e01 |
| * | 280 | 198.3 | 36.6 | 3.16e−03 | 4.2e01 |
| * | 320 | 171.8 | 15.0 | 2.07e−03 | 4.2e01 |
| * | 360 | 129.0 | 35.2 | 1.92e−03 | 4.2e01 |
|   | 400 | 286.0 | −35.0 | 2.76e−03 | 5.2e01 |
| - | 440 | 304.9 | −53.0 | 6.76e−02 | 1.0e02 |
| - | 480 | 307.8 | −44.9 | 6.33e−02 | 1.4e02 |

[Figure]

**Sample: 28B.1**

PCA  dec 163.43 / inc 22.83
PCA  MAD1 9.35 / MAD3 14.20
(−0.88 0.26 0.39)t

| | temp. | dec. | inc. | int. | m.s. |
|---|---|---|---|---|---|
| | 25 | 222.1 | 60.3 | 3.06e−02 | 9.4e01 |
| | 120 | 184.0 | 55.0 | 2.13e−02 | 9.2e01 |
| * | 160 | 170.1 | 41.8 | 1.82e−02 | 9.1e01 |
| * | 200 | 164.0 | 26.8 | 1.77e−02 | 9.0e01 |
| * | 240 | 162.0 | 24.9 | 1.70e−02 | 8.9e01 |
| * | 280 | 164.7 | 12.0 | 1.55e−02 | 8.7e01 |
| * | 320 | 159.2 | 0.4 | 1.27e−02 | 8.6e01 |
| * | 360 | 154.0 | 3.9 | 1.05e−02 | 8.8e01 |
| | 400 | 173.1 | −9.5 | 9.43e−03 | 9.4e01 |
| | 440 | 169.1 | −56.4 | 2.99e−02 | 1.2e02 |
| | 480 | 234.8 | −51.8 | 1.96e−02 | 1.4e02 |

[Figure]

**Sample: 29A.1**

□ vertical
■ horizontal
Units: A/m ×10$^{-2}$

PCA  dec 304.45 / inc 52.62
PCA  MAD1 10.83 / MAD3 6.42
(0.34 −0.50 0.79)t

| | temp. | dec. | inc. | int. | m.s. |
|---|---|---|---|---|---|
| * | 25 | 311.6 | 49.2 | 6.95e−02 | 1.2e02 |
| * | 120 | 303.0 | 55.6 | 4.71e−02 | 1.2e02 |
| * | 160 | 297.8 | 53.3 | 3.70e−02 | 1.1e02 |
| * | 200 | 294.1 | 55.0 | 3.09e−02 | 1.1e02 |
| * | 240 | 281.4 | 56.7 | 2.26e−02 | 1.1e02 |
| * | 280 | 273.6 | 57.2 | 1.56e−02 | 1.1e02 |
| | 320 | 266.1 | 63.9 | 1.42e−02 | 1.1e02 |
| | 360 | 253.2 | 59.2 | 1.06e−02 | 1.1e02 |
| | 400 | 201.6 | 67.6 | 8.20e−03 | 1.1e02 |
| | 440 | 213.6 | −59.6 | 6.50e−03 | 1.4e02 |
| | 480 | 247.6 | −41.4 | 1.02e−02 | 1.7e02 |

[Figure]

**Sample: 30A.1**

□ vertical
■ horizontal
 Units: A/m ×10$^{-3}$

PCA  dec 335.37 / inc 35.22
PCA  MAD1 16.66 / MAD3 5.81
(0.74 −0.34 0.58)t

| | temp. | dec. | inc. | int. | m.s. |
|---|---|---|---|---|---|
| * | 25 | 336.6 | 39.5 | 1.77e−02 | 2.7e01 |
| * | 120 | 335.4 | 34.9 | 1.08e−02 | 2.5e01 |
| * | 160 | 334.4 | 29.0 | 9.27e−03 | 2.6e01 |
| * | 200 | 329.9 | 36.1 | 6.83e−03 | 2.6e01 |
| * | 240 | 337.0 | 31.5 | 6.04e−03 | 2.5e01 |
| * | 280 | 331.3 | 25.4 | 4.73e−03 | 2.4e01 |
| * | 320 | 335.3 | 25.3 | 4.16e−03 | 2.3e01 |
| * | 360 | 332.1 | 11.9 | 3.47e−03 | 2.6e01 |
| * | 400 | 345.4 | 25.8 | 2.29e−03 | 3.5e01 |
| | 440 | 285.0 | −83.3 | 4.48e−03 | 8.4e01 |
| | 480 | 336.3 | −26.6 | 8.22e−03 | 1.2e02 |

[Figure]

**Sample: 31B**

□ vertical
■ horizontal
Units: A/m ×10$^{-3}$

PCA  dec 346.86 / inc 24.31
PCA  MAD1 16.39 / MAD3 23.09
(0.89 −0.21 0.41)t

| | temp. | dec. | inc. | int. | m.s. |
|---|---|---|---|---|---|
| * | 25 | 340.9 | 38.3 | 1.13e−02 | 3.5e01 |
| * | 120 | 334.5 | 33.1 | 6.45e−03 | 3.3e01 |
| * | 160 | 342.8 | 33.6 | 5.18e−03 | 3.4e01 |
| * | 200 | 4.3 | 47.1 | 5.04e−03 | 3.3e01 |
| * | 240 | 334.0 | 12.1 | 1.69e−03 | 3.2e01 |
| * | 280 | 357.4 | −8.9 | 1.04e−02 | 3.1e01 |
| * | 320 | 336.9 | 17.6 | 2.22e−03 | 3.2e01 |
| * | 360 | 341.8 | 10.7 | 2.09e−03 | 3.4e01 |
| | 400 | 37.4 | 1.7 | 2.01e−03 | 4.9e01 |
| | 440 | 91.5 | 2.6 | 8.52e−03 | 1.1e02 |
| - | 480 | 43.8 | 10.0 | 1.48e−02 | 2.0e02 |

[Figure]

**Sample: 32A.1**

□ vertical
■ horizontal
Units: A/m ×10$^{-3}$

PCA  dec 4.47 / inc 54.92
PCA  MAD1 13.22 / MAD3 2.79
(0.57 0.04 0.82)t

| | temp. | dec. | inc. | int. | m.s. |
|---|---|---|---|---|---|
| * | 25 | 360.0 | 55.7 | 1.52e−02 | 3.0e01 |
| * | 120 | 7.5 | 54.1 | 9.53e−03 | 3.8e01 |
| * | 160 | 6.5 | 54.4 | 7.72e−03 | 2.8e01 |
| * | 200 | 10.8 | 52.4 | 7.56e−03 | 2.8e01 |
| * | 240 | 10.5 | 55.6 | 6.28e−03 | 2.8e01 |
| | 280 | 322.6 | 25.6 | 3.84e−03 | 2.7e01 |
| | 320 | 355.8 | 53.7 | 4.73e−03 | 2.8e01 |
| | 360 | 14.8 | 38.6 | 3.77e−03 | 3.1e01 |
| | 400 | 359.3 | 57.0 | 3.07e−03 | 4.7e01 |
| | 440 | 340.8 | −39.9 | 7.08e−03 | 8.2e01 |
| - | 480 | 30.9 | 19.4 | 1.58e−02 | 1.7e02 |

[Figure]

**Sample: 33B**

□ vertical
■ horizontal
Units: A/m ×10$^{-3}$

PCA  dec 339.84 / inc 22.61
PCA  MAD1 29.56 / MAD3 8.78
(0.87 −0.32 0.38)t

| | temp. | dec. | inc. | int. | m.s. |
|---|---|---|---|---|---|
| * | 25 | 337.3 | 19.5 | 1.34e−02 | 2.8e01 |
| * | 120 | 343.3 | 22.7 | 7.42e−03 | 2.7e01 |
| * | 160 | 349.5 | 22.9 | 5.79e−03 | 2.6e01 |
| * | 200 | 345.5 | 37.5 | 4.43e−03 | 2.6e01 |
| * | 240 | 327.0 | 46.6 | 3.89e−03 | 2.6e01 |
| * | 280 | 339.7 | 3.2 | 2.79e−03 | 2.6e01 |
| * | 320 | 338.7 | 35.8 | 2.20e−03 | 2.6e01 |
| | 360 | 42.6 | −17.4 | 1.31e−03 | 2.9e01 |
| | 400 | 5.7 | −33.6 | 2.11e−03 | 4.6e01 |
| - | 440 | 65.7 | −30.2 | 1.24e−02 | 7.5e01 |
| - | 480 | 50.1 | −35.2 | 1.17e−02 | 1.3e02 |